# Analysis of Krylov Subspace Solutions of Regularized Nonconvex Quadratic Problems

**Yair Carmon**
Department of Electrical Engineering
Stanford University
yairc@stanford.edu

**John C. Duchi**
Departments of Statitstics and Electrical Engineering
Stanford University
jduchi@stanford.edu

## Abstract

We provide convergence rates for Krylov subspace solutions to the trust-region and cubic-regularized (nonconvex) quadratic problems. Such solutions may be efficiently computed by the Lanczos method and have long been used in practice. We prove error bounds of the form $1/t^2$ and $e^{-4t/\sqrt{\kappa}}$, where $\kappa$ is a condition number for the problem, and $t$ is the Krylov subspace order (number of Lanczos iterations). We also provide lower bounds showing that our analysis is sharp.

## 1 Introduction

Consider the potentially nonconvex quadratic function

$$f_{A,b}(x) := \frac{1}{2}x^T A x + b^T x,$$

where $A \in \mathbb{R}^{d \times d}$ and $b \in \mathbb{R}^d$. We wish to solve regularized minimization problems of the form

$$\underset{x}{\text{minimize}}\, f_{A,b}(x) \text{ subject to } \|x\| \leq R \quad \text{and} \quad \underset{x}{\text{minimize}}\, f_{A,b}(x) + \frac{\rho}{3}\|x\|^3, \qquad (1)$$

where $R$ and $\rho \geq 0$ are regularization parameters. These problems arise primarily in the family of trust-region and cubic-regularized Newton methods for general nonlinear optimization problems [11, 29, 18, 9], which optimize a smooth function $g$ by sequentially minimizing local models of the form

$$g(x_i + \Delta) \approx g(x_i) + \nabla g(x_i)^T \Delta + \frac{1}{2}\Delta^T \nabla^2 g(x_i)\Delta = g(x_i) + f_{\nabla^2 g(x_i), \nabla g(x_i)}(\Delta),$$

where $x_i$ is the current iterate and $\Delta \in \mathbb{R}^d$ is the search direction. Such models tend to be unreliable for large $\|\Delta\|$, particularly when $\nabla^2 g(x_i) \not\succ 0$. Trust-region and cubic regularization methods address this by constraining and regularizing the direction $\Delta$, respectively.

Both classes of methods and their associated subproblems are the subject of substantial ongoing research [19, 21, 5, 1, 25]. In the machine learning community, there is growing interest in using these methods for minimizing (often nonconvex) training losses, handling the large finite-sum structure of learning problems by means of sub-sampling [32, 23, 3, 38, 36].

The problems (1) are challenging to solve in high-dimensional settings, where direct decomposition (or even storage) of the matrix $A$ is infeasible. In some scenarios, however, computing matrix-vector products $v \mapsto Av$ is feasible. Such is the case when $A$ is the Hessian of a neural network, where $d$ may be in the millions and $A$ is dense, and yet we can compute Hessian-vector products efficiently on batches of training data [31, 33].

In this paper we consider a scalable approach for approximately solving (1), which consists of minimizing the objective in the *Krylov subspace* of order $t$,

$$\mathcal{K}_t(A, b) := \text{span}\{b, Ab, \ldots, A^{t-1}b\}. \qquad (2)$$

This requires only $t$ matrix-vector products, and the Lanczos method allows one to efficiently find the solution to problems (1) over $\mathcal{K}_t(A, b)$ (see, e.g. [17, 9, Sec. 2]). Krylov subspace methods are familiar in numerous large-scale numerical problems, including conjugate gradient methods, eigenvector problems, or solving linear systems [20, 26, 35, 14].

It is well-known that, with exact arithmetic, the order $d$ subspace $\mathcal{K}_d(A, b)$ generically contains the global solutions to (1). However, until recently the literature contained no guarantees on the rate at which the suboptimality of the solution approaches zero as the subspace dimension $t$ grows. This is in contrast to the two predominant Krylov subspace method use-cases—convex quadratic optimization [14, 27, 28] and eigenvector finding [24]—where such rates of convergence have been known for decades. Zhang et al. [39] make substantial progress on this gap, establishing bounds implying a linear rate of convergence for the trust-region variant of problem (1).

In this work we complete the picture, proving that the optimality gap of the order $t$ Krylov subspace solution to either of the problems (1) is bounded by both $e^{-4t/\sqrt{\kappa}}$ and $t^{-2}\log^2(\|b\|/|u_{\min}^T b|)$. Here $\kappa$ is a condition number for the problem that naturally generalizes the classical condition number of the matrix $A$, and $u_{\min}$ is an eigenvector of $A$ corresponding to its smallest eigenvalue. Using randomization, we may replace $|u_{\min}^T b|$ with a term proportional to $1/\sqrt{d}$, circumventing the well-known "hard case" of the problem (1) (see Section 2.5). Our analysis both leverages and unifies the known results for convex quadratic and eigenvector problems, which constitute special cases of (1).

**Related work** Zhang et al. [39] show that the error of certain polynomial approximation problems bounds the suboptimality of Krylov subspace solutions to the trust region-variant of the problems (1), implying convergence at a rate exponential in $-t/\sqrt{\kappa}$. Based on these bounds, the authors propose novel stopping criteria for subproblem solutions in the trust-region optimization method, showing good empirical results. However, the bounds of [39] become weak for large $\kappa$ and vacuous in the hard case where $\kappa = \infty$.

Prior works develop algorithms for solving (1) with convergence guarantees that hold in the hard case. Hazan and Koren [19], Ho-Nguyen and Kılınç-Karzan [21], and Agarwal et al. [1] propose algorithms that obtain error roughly $t^{-2}$ after computing $t$ matrix-vector products. The different algorithms these papers propose all essentially reduce the problems (1) to a sequence of eigenvector and convex quadratic problems to which standard algorithms apply. In previous work [5], we analyze gradient descent—a direct, local method—for the cubic-regularized problem. There, we show a rate of convergence roughly $t^{-1}$, reflecting the well-known complexity gap between gradient descent (respectively, the power method) and conjugate gradient (respectively, Lanczos) methods [35, 14].

Our development differs from this prior work in the following ways.

1. We analyze a practical approach, implemented in efficient optimization libraries [16, 25], with essentially no tuning parameters. Previous algorithms [19, 21, 1] are convenient for theoretical analysis but less conducive to efficient implementation; each has several parameters that require tuning, and we are unaware of numerical experiments with any of the approaches.

2. We provide both linear ($e^{-4t/\sqrt{\kappa}}$) and sublinear ($t^{-2}$) convergence guarantees. In contrast, the papers [19, 21, 1] provide only a sublinear rate; Zhang et al. [39] provide only the linear rate.

3. Our analysis applies to both the trust-region and cubic regularization variants in (1), while [19, 21, 39] consider only the trust-region problem, and [39, 5] consider only cubic regularization.

4. We provide lower bounds—for adversarially constructed problem instances—showing our convergence guarantees are tight to within numerical constants. By a resisting oracle argument [27], these bounds apply to any deterministic algorithm that accesses $A$ via matrix-vector products.

5. Our arguments are simple and transparent, and we leverage established results on convex optimization and the eigenvector problem to give short proofs of our main results.

**Paper organization** In Section 2 we state and prove our convergence rate guarantees for the trust-region problem. Then, in Section 3 we quickly transfer those results to the cubic-regularized problem by showing that it always has a smaller optimality gap. Section 4 gives our lower bounds, stated for cubic regularization but immediately applicable to the trust-region problem by the same optimality gap bound. Finally, in Section 5 we illustrate our analysis with some numerical experiments.

**Notation** For a symmetric matrix $A \in \mathbb{R}^{d \times d}$ and vector $b$ we let $f_{A,b}(x) := \frac{1}{2}x^T A x + b^T x$. We let $\lambda_{\min}(A)$ and $\lambda_{\max}(A)$ denote the minimum and maximum eigenvalues of $A$, and let $u_{\min}(A), u_{\max}(A)$ denote their corresponding (unit) eigenvectors, dropping the argument $A$ when clear from context. For integer $t \geq 1$ we let $\mathcal{P}_t := \{c_0 + c_1 x + \cdots + c_{t-1}x^{t-1} \mid c_i \in \mathbb{R}\}$ be the polynomials of degree at most $t-1$, so that the Krylov subspace (2) is $\mathcal{K}_t(A, b) = \{p(A)b \mid p \in \mathcal{P}_t\}$. We use $\|\cdot\|$ to denote Euclidean norm on $\mathbb{R}^d$ and $\ell_2$-operator norm on $\mathbb{R}^{d \times d}$. Finally, we denote $(z)_+ := \max\{z, 0\}$ and $(z)_- := \min\{z, 0\}$.

## 2 The trust-region problem

Fixing a symmetric matrix $A \in \mathbb{R}^{d \times d}$, vector $b \in \mathbb{R}^d$ and trust-region radius $R > 0$, we let

$$s_\star^{\mathsf{tr}} \in \operatorname*{argmin}_{x \in \mathbb{R}^d, \, \|x\| \leq R} f_{A,b}(x) = \frac{1}{2}x^T A x + b^T x$$

denote a solution (global minimizer) of the trust region problem. Letting $\lambda_{\min}, \lambda_{\max}$ denote the extremal eigenvalues of $A$, $s_\star^{\mathsf{tr}}$ admits the following characterization [11, Ch. 7]: $s_\star^{\mathsf{tr}}$ solves problem (1) if and only if there exists $\lambda_\star$ such that

$$(A + \lambda_\star I)s_\star^{\mathsf{tr}} = -b, \quad \lambda_\star \geq (-\lambda_{\min})_+, \quad \text{and} \quad \lambda_\star(R - \|s_\star^{\mathsf{tr}}\|) = 0. \tag{3}$$

The optimal Lagrange multiplier $\lambda_\star$ always exists and is unique, and if $\lambda_\star > -\lambda_{\min}$ the solution $s_\star^{\mathsf{tr}}$ is unique and satisfies $s_\star^{\mathsf{tr}} = -(A + \lambda_\star I)^{-1}b$. Letting $u_{\min}$ denote the eigenvector of $A$ corresponding to $\lambda_{\min}$, the characterization (3) shows that $u_{\min}^T b \neq 0$ implies $\lambda_\star > -\lambda_{\min}$.

Now, consider the Krylov subspace solutions, and for $t > 0$, let

$$s_t^{\mathsf{tr}} \in \operatorname*{argmin}_{x \in \mathcal{K}_t(A,b), \, \|x\| \leq R} f_{A,b}(x) = \frac{1}{2}x^T A x + b^T x$$

denote a minimizer of the trust-region problem in the Krylov subspace of order $t$. Gould et al. [17] show how to compute the Krylov subspace solution $s_t^{\mathsf{tr}}$ in time dominated by the cost of computing $t$ matrix-vector products using the Lanczos method (see also Section A of the supplement).

### 2.1 Main result

With the notation established above, our main result follows.

**Theorem 1.** *For every* $t > 0$,

$$f_{A,b}(s_t^{\mathsf{tr}}) - f_{A,b}(s_\star^{\mathsf{tr}}) \leq 36 \left[ f_{A,b}(0) - f_{A,b}(s_\star^{\mathsf{tr}}) \right] \exp\left\{ -4t\sqrt{\frac{\lambda_{\min} + \lambda_\star}{\lambda_{\max} + \lambda_\star}} \right\}, \tag{4}$$

*and*

$$f_{A,b}(s_t^{\mathsf{tr}}) - f_{A,b}(s_\star^{\mathsf{tr}}) \leq \frac{(\lambda_{\max} - \lambda_{\min}) \|s_\star^{\mathsf{tr}}\|^2}{(t - \frac{1}{2})^2} \left[ 4 + \frac{\mathbb{I}_{\{\lambda_{\min} < 0\}}}{8} \log^2 \left( \frac{4\|b\|^2}{(u_{\min}^T b)^2} \right) \right]. \tag{5}$$

Theorem 1 characterizes two convergence regimes: linear (4) and sublinear (5). Linear convergence occurs when $t \gtrsim \sqrt{k}$, where $\kappa = \frac{\lambda_{\max} + \lambda_\star}{\lambda_{\min} + \lambda_\star} \geq 1$ is the condition number for the problem. There, the error decays exponentially and falls beneath $\epsilon$ in roughly $\sqrt{\kappa} \log \frac{1}{\epsilon}$ Lanczos iteration. Sublinear convergence occurs when $t \lesssim \sqrt{k}$, and there the error decays polynomially and falls beneath $\epsilon$ in roughly $\frac{1}{\sqrt{\epsilon}}$ iterations. For worst-case problem instances this characterization is tight to constant factors, as we show in Section 4.

The guarantees of Theorem 1 closely resemble the well-known guarantees for the conjugate gradient method [35], including them as the special case $R = \infty$ and $\lambda_{\min} \geq 0$. For convex problems, the radius constraint $\|x\| \leq R$ always improves the conditioning of the problem, as $\frac{\lambda_{\max}}{\lambda_{\min}} \geq \frac{\lambda_{\max} + \lambda_\star}{\lambda_{\min} + \lambda_\star}$; the smaller $R$ is, the better conditioned the problem becomes. For non-convex problems, the sublinear rate features an additional logarithmic term that captures the role of the eigenvector $u_{\min}$. The

first rate (4) is similar to those of Zhang et al. [39, Thm. 4.11], though with somewhat more explicit dependence on $t$.

In the "hard case," which corresponds to $u_{\min}^T b = 0$ and $\lambda_{\min} + \lambda_\star = 0$ (cf. [11, Ch. 7]), both the bounds in Theorem 1 become vacuous, and indeed $s_t^{\mathsf{tr}}$ may not converge to the global minimizer in this case. However, as the bound (5) depends only logarithmically on $u_{\min}^T b$, it remains valid even extremely close to the hard case. In Section 2.5 we describe two simple randomization techniques with convergence guarantees that are valid in the hard case as well.

## 2.2 Proof sketch

Our analysis reposes on two elementary observations. First, we note that Krylov subspaces are invariant to shifts by scalar matrices, *i.e.* $\mathcal{K}_t(A, b) = \mathcal{K}_t(A_\lambda, b)$ for any $A, b, t$ where $\lambda \in \mathbb{R}$, and

$$A_\lambda := A + \lambda I.$$

Second, we observe that for every point $x$ and $\lambda \in \mathbb{R}$

$$f_{A,b}(x) - f_{A,b}(s_\star^{\mathsf{tr}}) = f_{A_\lambda,b}(x) - f_{A_\lambda,b}(s_\star^{\mathsf{tr}}) + \frac{\lambda}{2}(\left\|s_\star^{\mathsf{tr}}\right\|^2 - \|x\|^2) \tag{6}$$

Our strategy then is to choose $\lambda$ such that $A_\lambda \succeq 0$, and then use known results to find $y_t \in \mathcal{K}_t(A_\lambda, b) = \mathcal{K}_t(A, b)$ that rapidly reduces the "convex error" term $f_{A_\lambda,b}(y_t) - f_{A_\lambda,b}(s_\star^{\mathsf{tr}})$. We then adjust $y_t$ to obtain a feasible point $x_t$ such that the "norm error" term $\frac{\lambda}{2}(\left\|s_\star^{\mathsf{tr}}\right\|^2 - \|x_t\|^2)$ is small. To establish linear convergence, we take $\lambda = \lambda_\star$ and adjust the norm of $y_t$ by taking $x_t = (1-\alpha)y_t$ for some small $\alpha$ that guarantees $x_t$ is feasible and that the "norm error" term is small. To establish sublinear convergence we set $\lambda = -\lambda_{\min}$ and take $x_t = y_t + \alpha \cdot z_t$, where $z_t$ is an approximation for $u_{\min}$ within $\mathcal{K}_t(A, b)$, and $\alpha$ is chosen to make $\|x_t\| = \|s_\star^{\mathsf{tr}}\|$. This means the "norm error" vanishes, while the "convex error" cannot increase too much, as $A_{-\lambda_{\min}} z_t \approx A_{-\lambda_{\min}} u_{\min} = 0$.

Our approach for proving the sublinear rate of convergence is inspired by Ho-Nguyen and Kılınç-Karzan [21], who also rely on Nesterov's method in conjunction with Lanczos-based eigenvector approximation. The analysis in [21] uses an algorithmic reduction, proposing to apply the Lanczos method (with a random vector instead of $b$) to approximate $u_{\min}$ and $\lambda_{\min}$, then run Nesterov's method on an approximate version of the "convex error" term, and then use the approximated eigenvector to adjust the norm of the result. We instead argue that all the ingredients for this reduction already exist in the Krylov subspace $\mathcal{K}_t(A, b)$, obviating the need for explicit eigenvector estimation or actual application of accelerated gradient descent.

## 2.3 Building blocks

Our proof uses the following classical results.

**Lemma 1** (Approximate matrix inverse). *Let* $\alpha, \beta$ *satisfy* $0 < \alpha \leq \beta$, *and let* $\kappa = \beta/\alpha$. *For any* $t \geq 1$ *there exists a polynomial* $p$ *of degree at most* $t - 1$, *such that for every* $M$ *satisfying* $\alpha I \preceq M \preceq \beta I$,

$$\|I - Mp(M)\| \leq 2e^{-2t/\sqrt{\kappa}}.$$

**Lemma 2** (Convex trust-region problem). *Let* $t \geq 1$, $M \succeq 0$, $v \in \mathbb{R}^d$ *and* $r \geq 0$, *and let* $f_{M,v}(x) = \frac{1}{2}x^T M x + v^T x$. *There exists* $x_t \in \mathcal{K}_t(M, v)$ *such that*

$$\|x_t\| \leq r \quad and \quad f_{M,v}(x_t) - \min_{\|x\|\leq r} f_{M,v}(x) \leq \frac{4\lambda_{\max}(M) \cdot r^2}{(t+1)^2}.$$

**Lemma 3** (Finding eigenvectors, [24, Theorem 4.2]). *Let* $M \succeq 0$ *be such that* $u^T M u = 0$ *for some unit vector* $u \in \mathbb{R}^d$, *and let* $v \in \mathbb{R}^d$. *For every* $t \geq 1$ *there exists* $z_t \in \mathcal{K}_t(M, v)$ *such that*

$$\|z_t\| = 1 \quad and \quad z_t^T M z_t \leq \frac{\|M\|}{16(t - \frac{1}{2})^2} \log^2\left(-2 + 4\frac{\|v\|^2}{(u^T v)^2}\right).$$

While these lemmas are standard, their explicit forms are useful, and we prove them in Section C.1 in the supplement. Lemmas 1 and 3 are consequences of uniform polynomial approximation results (cf. supplement, Sec. B). To prove Lemma 2 we invoke Tseng's results on a variant of Nesterov's accelerated gradient method [37], arguing that its iterates lie in the Krylov subspace.

## 2.4 Proof of Theorem 1

**Linear convergence**  Recalling the notation $A_{\lambda_\star} = A + \lambda_\star I$, let $y_t = -p(A_{\lambda_\star})b = p(A_{\lambda_\star})A_{\lambda_\star}s_\star^{\mathrm{tr}}$, for the $p \in \mathcal{P}_t$ which Lemma 1 guarantees to satisfy $\|p(A_{\lambda_\star})A_{\lambda_\star} - I\| \le 2e^{-2t/\sqrt{\kappa(A_{\lambda_\star})}}$. Let

$$x_t = (1-\alpha)y_t, \text{ where } \alpha = \frac{\|y_t\| - \|s_\star^{\mathrm{tr}}\|}{\max\{\|s_\star^{\mathrm{tr}}\|, \|y_t\|\}},$$

so that we are guaranteed $\|x_t\| \le \|s_\star^{\mathrm{tr}}\|$ for any value of $\|y_t\|$. Moreover

$$|\alpha| = \frac{|\,\|y_t\| - \|s_\star^{\mathrm{tr}}\|\,|}{\max\{\|s_\star^{\mathrm{tr}}\|, \|y_t\|\}} \le \frac{\|y_t - s_\star^{\mathrm{tr}}\|}{\|s_\star^{\mathrm{tr}}\|} = \frac{\|(p(A_{\lambda_\star})A_{\lambda_\star} - I)s_\star^{\mathrm{tr}}\|}{\|s_\star^{\mathrm{tr}}\|} \le 2e^{-2t/\sqrt{\kappa(A_{\lambda_\star})}},$$

where the last transition used $\|p(A_{\lambda_\star})A_{\lambda_\star} - I\| \le 2e^{-2t/\sqrt{\kappa(A_{\lambda_\star})}}$.

Since $b = -A_{\lambda_\star}s_\star^{\mathrm{tr}}$, we have $f_{A_{\lambda_\star},b}(x) = f_{A_{\lambda_\star},b}(s_\star^{\mathrm{tr}}) + \frac{1}{2}\|A_{\lambda_\star}^{1/2}(x - s_\star^{\mathrm{tr}})\|^2$. The equality (6) with $\lambda = \lambda_\star$ and $\|x_t\| \le \|s_\star^{\mathrm{tr}}\|$ therefore implies

$$f_{A,b}(x_t) - f_{A,b}(s_\star^{\mathrm{tr}}) \le \frac{1}{2}\left\|A_{\lambda_\star}^{1/2}(x_t - s_\star^{\mathrm{tr}})\right\|^2 + \lambda_\star\left\|s_\star^{\mathrm{tr}}\right\|(\left\|s_\star^{\mathrm{tr}}\right\| - \|x_t\|). \tag{7}$$

When $\|y_t\| \ge \|s_\star^{\mathrm{tr}}\|$ we have $\|x_t\| = \|s_\star^{\mathrm{tr}}\|$ and the second term vanishes. When $\|y_t\| < \|s_\star^{\mathrm{tr}}\|$,

$$\left\|s_\star^{\mathrm{tr}}\right\| - \|x_t\| = \left\|s_\star^{\mathrm{tr}}\right\| - \|y_t\| - \frac{\|y_t\|}{\|s_\star^{\mathrm{tr}}\|} \cdot (\left\|s_\star^{\mathrm{tr}}\right\| - \|y_t\|) = \left\|s_\star^{\mathrm{tr}}\right\|\alpha^2 \le 4e^{-4t/\sqrt{\kappa(A_{\lambda_\star})}}\left\|s_\star^{\mathrm{tr}}\right\|. \tag{8}$$

We also have,

$$\left\|A_{\lambda_\star}^{1/2}(x_t - s_\star^{\mathrm{tr}})\right\| = \left\|([1-\alpha]p(A_{\lambda_\star})A_{\lambda_\star} - I)A_{\lambda_\star}^{1/2}s_\star^{\mathrm{tr}}\right\|$$
$$\le (1 + |\alpha|)\left\|(p(A_{\lambda_\star})A_{\lambda_\star} - I)A_{\lambda_\star}^{1/2}s_\star^{\mathrm{tr}}\right\| + |\alpha|\left\|A_{\lambda_\star}^{1/2}s_\star^{\mathrm{tr}}\right\| \le 6\left\|A_{\lambda_\star}^{1/2}s_\star^{\mathrm{tr}}\right\|e^{-2t/\sqrt{\kappa(A_{\lambda_\star})}}, \tag{9}$$

where in the final transition we used our upper bounds on $\alpha$ and $\|p(A_{\lambda_\star})A_{\lambda_\star} - I\|$, as well as $|\alpha| \le 1$. Substituting the bounds (8) and (9) into inequality (7), we have

$$f_{A,b}(x_t) - f_{A,b}(s_\star^{\mathrm{tr}}) \le \left(18s_\star^{\mathrm{tr}\,T}A_{\lambda_\star}s_\star^{\mathrm{tr}} + 4\lambda_\star\left\|s_\star^{\mathrm{tr}}\right\|^2\right)e^{-4t/\sqrt{\kappa(A_{\lambda_\star})}}, \tag{10}$$

and the final bound follows from recalling that $f_{A,b}(0) - f_{A,b}(s_\star^{\mathrm{tr}}) = \frac{1}{2}s_\star^{\mathrm{tr}\,T}A_{\lambda_\star}s_\star^{\mathrm{tr}} + \frac{\lambda_\star}{2}\|s_\star^{\mathrm{tr}}\|^2$ and substituting $\kappa(A_{\lambda_\star}) = (\lambda_{\max} + \lambda_\star)/(\lambda_{\min} + \lambda_\star)$. To conclude the proof we note that $(1 - \alpha)p(A_{\lambda_\star}) = (1 - \alpha)p(A + \lambda_\star I) = \tilde{p}(A)$ for some $\tilde{p} \in \mathcal{P}_t$, so that $x_t \in \mathcal{K}_t(A, b)$ and $\|x_t\| \le R$, and therefore $f_{A,b}(s_t^{\mathrm{tr}}) \le f_{A,b}(x_t)$.

**Sublinear convergence**  Let $A_0 := A - \lambda_{\min}I \succeq 0$ and apply Lemma 2 with $M = A_0$, $v = b$ and $r = \|s_\star^{\mathrm{tr}}\|$ to obtain $y_t \in \mathcal{K}_t(A_0, b) = \mathcal{K}_t(A, b)$ such that

$$\|y_t\| \le \left\|s_\star^{\mathrm{tr}}\right\| \text{ and } f_{A_0,b}(y_t) - f_{A_0,b}(s_\star^{\mathrm{tr}}) \le f_{A_0,b}(y_t) - \min_{\|x\| \le \|s_\star^{\mathrm{tr}}\|} f_{A_0,b}(x) \le \frac{4\|A_0\|\|s_\star^{\mathrm{tr}}\|^2}{(t+1)^2}. \tag{11}$$

If $\lambda_{\min} \ge 0$, equality (6) with $\lambda = -\lambda_{\min}$ along with (11) means we are done, recalling that $\|A_0\| = \lambda_{\max} - \lambda_{\min}$. For $\lambda_{\min} < 0$, apply Lemma 3 with $M = A_0$ and $v = b$ to obtain $z_t \in \mathcal{K}_t(A, b)$ such that

$$\|z_t\| = 1 \text{ and } z_t^T A_0 z_t \le \frac{\|A_0\|}{16(t - \frac{1}{2})^2}\log^2\left(4\frac{\|b\|^2}{(u_{\min}^T b)^2}\right). \tag{12}$$

We form the vector

$$x_t = y_t + \alpha \cdot z_t \in \mathcal{K}_t(A, b),$$

and choose $\alpha$ to satisfy

$$\|x_t\| = \left\|s_\star^{\mathrm{tr}}\right\| \text{ and } \alpha \cdot z_t^T(A_0 y_t + b) = \alpha \cdot z_t^T \nabla f_{A_0,b}(y_t) \le 0.$$

We may always choose such $\alpha$ because $\|y_t\| \le \|s_\star^{\mathrm{tr}}\|$ and therefore $\|y_t + \alpha z_t\| = \|s_\star^{\mathrm{tr}}\|$ has both a non-positive and a non-negative solution in $\alpha$. Moreover because $\|z_t\| = 1$ we have that $|\alpha| \le$

$2 \| s_\star^{\mathsf{tr}} \|$. The property $\alpha \cdot z_t^T \nabla f_{A_0, b}(y_t) \leq 0$ of our construction of $\alpha$ along with $\nabla^2 f_{A_0, b} = A_0$, gives us,

$$f_{A_0, b}(x_t) = f_{A_0, b}(y_t) + \alpha \cdot z_t^T \nabla f_{A_0, b}(y_t) + \frac{\alpha^2}{2} z_t^T A_0 z_t \leq f_{A_0, b}(y_t) + \frac{\alpha^2}{2} z_t^T A_0 z_t.$$

Substituting this bound along with $\|x_t\| = \| s_\star^{\mathsf{tr}} \|$ and $\alpha^2 \leq 4 \| s_\star^{\mathsf{tr}} \|^2$ into (6) with $\lambda = -\lambda_{\min}$ gives

$$f_{A, b}(x_t) - f_{A, b}(s_\star^{\mathsf{tr}}) \leq f_{A_0, b}(y_t) - f_{A_0, b}(s_\star^{\mathsf{tr}}) + 2 \| s_\star^{\mathsf{tr}} \|^2 z_t^T A_0 z_t.$$

Substituting in the bounds (11) and (12) concludes the proof for the case $\lambda_{\min} < 0$.

## 2.5 Randomizing away the hard case

Krylov subspace solutions may fail to converge to global solution when both $\lambda_\star = -\lambda_{\min}$ and $u_{\min}^T b = 0$, the so-called hard case [11, 30]. Yet as with eigenvector methods [24, 14], simple randomization approaches allow us to handle the hard case with high probability, at the modest cost of introducing to the error bounds a logarithmic dependence on $d$. Here we describe two such approaches.

In the first approach, we draw a spherically symmetric random vector $v$, and consider the *joint Krylov subspace*

$$\mathcal{K}_{2t}(A, \{b, v\}) := \mathrm{span}\{b, Ab, \dots, A^{t-1} b, v, Av, \dots, A^{t-1} v\}.$$

The trust-region and cubic-regularized problems (1) can be solved efficiently in $\mathcal{K}_{2t}(A, \{b, v\})$ using the *block Lanczos* method [12, 15]; we survey this technique in Section A.1 in the supplement. The analysis in the previous section immediately implies the following convergence guarantee.

**Corollary 2.** *Let $v$ be uniformly distributed on the unit sphere in $\mathbb{R}^d$, and*

$$\hat{s}_t^{\mathsf{tr}} \in \underset{x \in \mathcal{K}_{\lfloor t/2 \rfloor}(A, \{b, v\}), \|x\| \leq R}{\mathrm{argmin}} f_{A, b}(x).$$

*For any $\delta > 0$,*

$$f_{A, b}(\hat{s}_t^{\mathsf{tr}}) - f_{A, b}(s_\star^{\mathsf{tr}}) \leq \frac{(\lambda_{\max} - \lambda_{\min}) R^2}{(t-1)^2} \left[ 16 + 2 \cdot \mathbb{I}_{\{\lambda_{\min} < 0\}} \log^2 \left( \frac{2\sqrt{d}}{\delta} \right) \right] \qquad (13)$$

*with probability at least $1 - \delta$ with respect to the random choice of $v$.*

*Proof.* In the preceding proof of sublinear convergence, apply Lemma 2 on $\mathcal{K}_{\lfloor t/2 \rfloor}(A, b)$ and Lemma 3 on $\mathcal{K}_{\lfloor t/2 \rfloor}(A, v)$; the constructed solution is in $\mathcal{K}_{\lfloor t/2 \rfloor}(A, \{b, v\})$. To bound $|u_{\min}^T v|^2 / \|v\|^2$, note that its distribution is Beta$(\frac{1}{2}, \frac{d-1}{2})$ and therefore $|u_{\min}^T v|^2 / \|v\|^2 \geq \delta^2 / d$ with probability greater than $1 - \delta$ (cf. [5, Lemma 4.6]). $\qquad \square$

Corollary 2 implies we can solve the trust-region problem to $\epsilon$ accuracy in roughly $\epsilon^{-1/2} \log d$ matrix-vector products, even in the hard case. The main drawback of this randomization approach is that half the matrix-vector products are expended on the random vector; when the problem is well-conditioned or when $|u_{\min}^T b| / \|b\|$ is not extremely small, using the standard subspace solution is nearly twice as fast.

The second approach follows the proposal [5] to construct a perturbed version of the linear term $b$, denoted $\tilde{b}$, and solve the problem instance $(A, \tilde{b}, R)$ in the Krylov subspace $\mathcal{K}_t(A, \tilde{b})$.

**Corollary 3.** *Let $v$ be uniformly distributed on the unit sphere in $\mathbb{R}^d$, let $\sigma > 0$ and let*

$$\tilde{b} = b + \sigma \cdot v.$$

*Let $\tilde{s}_t^{\mathsf{tr}} \in \mathrm{argmin}_{x \in \mathcal{K}_t(A, \tilde{b}), \|x\| \leq R} f_{A, \tilde{b}}(x) := \frac{1}{2} x^T A x + \tilde{b}^T x$. For any $\delta > 0$,*

$$f_{A, b}(\tilde{s}_t^{\mathsf{tr}}) - f_{A, b}(s_\star^{\mathsf{tr}}) \leq \frac{(\lambda_{\max} - \lambda_{\min}) R^2}{(t - \frac{1}{2})^2} \left[ 4 + \frac{\mathbb{I}_{\{\lambda_{\min} < 0\}}}{2} \log^2 \left( \frac{2 \| \tilde{b} \| \sqrt{d}}{\sigma \delta} \right) \right] + 2\sigma R \qquad (14)$$

*with probability at least $1 - \delta$ with respect to the random choice of $v$.*

See section C.2 in the supplement for a short proof, which consists of arguing that $f_{A,b}$ and $f_{A,\tilde{b}}$ deviate by at most $\sigma R$ at any feasible point, and applying a probabilistic lower bound on $|u_{\min}^T \tilde{b}|$. For any desired accuracy $\epsilon$, using Corollary 3 with $\sigma = \epsilon/(4R)$ shows we can achieve this accuracy, with constant probability, in a number of Lanczos iterations that scales as $\epsilon^{-1/2} \log(d/\epsilon^2)$. Compared to the first approach, this rate of convergence is asymptotically slightly slower (by a factor of $\log \frac{1}{\epsilon}$), and moreover requires us to decide on a desired level of accuracy in advance. However, the second approach avoids the 2x slowdown that the first approach exhibits on easier problem instances. In Section 5 we compare the two approaches empirically.

We remark that the linear convergence guarantee (4) continues to hold for both randomization approaches. For the second approach, this is due to the fact that small perturbations to $b$ do not drastically change the condition number, as shown in [5]. However, this also means that we cannot expect a good condition number when perturbing $b$ in the hard case. Nevertheless, we believe it is possible to show that, with randomization, Krylov subspace methods exhibit linear convergence even in the hard case, where the condition number is replaced by the normalized eigen-gap $(\lambda_{\max} - \lambda_{\min})/(\lambda_2 - \lambda_{\min})$, with $\lambda_2$ the smallest eigenvalue of $A$ larger than $\lambda_{\min}$.

## 3 The cubic-regularized problem

We now consider the cubic-regularized problem

$$\underset{x\in\mathbb{R}^d}{\text{minimize}} \ \hat{f}_{A,b,\rho}(x) := f_{A,b}(x) + \frac{\rho}{3}\|x\|^3 = \frac{1}{2}x^T A x + b^T x + \frac{\rho}{3}\|x\|^3.$$

Any global minimizer of $\hat{f}_{A,b,\rho}$, denoted $s_\star^{\text{cr}}$, admits the characterization [9, Theorem 3.1]

$$\nabla \hat{f}_{A,b,\rho}(s_\star^{\text{cr}}) = \left(A + \rho\|s_\star^{\text{cr}}\| I\right) s_\star^{\text{cr}} + b = 0 \ \text{ and } \ \rho\|s_\star^{\text{cr}}\| \geq -\lambda_{\min}. \tag{15}$$

Comparing this characterization to its counterpart (3) for the trust-region problem, we see that any instance $(A, b, \rho)$ of cubic regularization has an *equivalent trust-region instance* $(A, b, R)$, with $R = \|s_\star^{\text{cr}}\|$. Theses instances are equivalent in that they have the same set of global minimizers. Evidently, the equivalent trust-region instance has optimal Lagrange multiplier $\lambda_\star = \rho\|s_\star^{\text{cr}}\|$. Moreover, at any trust-region feasible point $x$ (satisfying $\|x\| \leq R = \|s_\star^{\text{cr}}\| = \|s_\star^{\text{tr}}\|$), the cubic-regularization optimality gap is smaller than its trust-region equivalent,

$$\hat{f}_{A,b,\rho}(x) - \hat{f}_{A,b,\rho}(s_\star^{\text{cr}}) = f_{A,b}(x) - f_{A,b}(s_\star^{\text{tr}}) + \frac{\rho}{3}\left(\|x\|^3 - \|s_\star^{\text{tr}}\|^3\right) \leq f_{A,b}(x) - f_{A,b}(s_\star^{\text{tr}}).$$

Letting $s_t^{\text{cr}}$ denote the minimizer of $\hat{f}_{A,b,\rho}$ in $\mathcal{K}_t(A, b)$ and letting $s_t^{\text{tr}}$ denote the Krylov subspace solution of the equivalent trust-region problem, we conclude that

$$\hat{f}_{A,b,\rho}(s_t^{\text{cr}}) - \hat{f}_{A,b,\rho}(s_\star^{\text{cr}}) \leq \hat{f}_{A,b,\rho}(s_t^{\text{tr}}) - \hat{f}_{A,b,\rho}(s_\star^{\text{cr}}) \leq f_{A,b}(s_t^{\text{tr}}) - f_{A,b}(s_\star^{\text{tr}}); \tag{16}$$

cubic regularization Krylov subspace solutions always have a *smaller optimality gap* than their trust-region equivalents. The guarantees of Theorem 1 therefore apply to $\hat{f}_{A,b,\rho}(s_t^{\text{cr}}) - \hat{f}_{A,b,\rho}(s_\star^{\text{cr}})$ as well, and we arrive at the following

**Corollary 4.** *For every $t > 0$,*

$$\hat{f}_{A,b,\rho}(s_t^{\text{cr}}) - \hat{f}_{A,b,\rho}(s_\star^{\text{cr}}) \leq 36 \left[\hat{f}_{A,b,\rho}(0) - \hat{f}_{A,b,\rho}(s_\star^{\text{cr}})\right] \exp\left\{-4t\sqrt{\frac{\lambda_{\min} + \rho\|s_\star^{\text{cr}}\|}{\lambda_{\max} + \rho\|s_\star^{\text{cr}}\|}}\right\}, \tag{17}$$

*and*

$$\hat{f}_{A,b,\rho}(s_t^{\text{cr}}) - \hat{f}_{A,b,\rho}(s_\star^{\text{cr}}) \leq \frac{(\lambda_{\max} - \lambda_{\min})\|s_\star^{\text{cr}}\|^2}{(t - \frac{1}{2})^2}\left[4 + \frac{\mathbb{I}_{\{\lambda_{\min}<0\}}}{8}\log^2\left(\frac{4\|b\|^2}{(u_{\min}^T b)^2}\right)\right]. \tag{18}$$

*Proof.* Use the slightly stronger bound (10) derived in the proof of Theorem 1 with the inequality $18 s_\star^{\text{tr}T} A_{\lambda_\star} s_\star^{\text{tr}} + 4\lambda_\star\|s_\star^{\text{tr}}\|^2 \leq 36[\frac{1}{2}s_\star^{\text{cr}T} A s_\star^{\text{cr}} + \frac{1}{6}\rho\|s_\star^{\text{cr}}\|^3] = 36[\hat{f}_{A,b,\rho}(0) - \hat{f}_{A,b,\rho}(s_\star^{\text{cr}})]$. $\qquad\square$

Here too it is possible to randomly perturb $b$ and obtain a guarantee for cubic regularization that applies in the hard case. In [5] we carry out such analysis for gradient descent, and show that perturbations to $b$ with norm $\sigma$ can increase $\|s_\star^{cr}\|^2$ by at most $2\sigma/\rho$ [5, Lemma 4.6]. Thus the cubic-regularization equivalent of Corollary 3 amounts to replacing $R^2$ with $\|s_\star^{cr}\|^2 + 2\sigma/\rho$ in (14).

We note briefly—without giving a full analysis—that Corollary 4 shows that the practically successful Adaptive Regularization using Cubics (ARC) method [9] can find $\epsilon$-stationary points in roughly $\epsilon^{-7/4}$ Hessian-vector product operations (with proper randomization and subproblem stopping criteria). Researchers have given such guarantees for a number of algorithms that are mainly theoretical [1, 8], as well as variants of accelerated gradient descent [6, 22], which while more practical still require careful parameter tuning. In contrast, ARC requires very little tuning and it is encouraging that it may also exhibit the enhanced Hessian-vector product complexity $\epsilon^{-7/4}$, which is at least near-optimal [7].

## 4 Lower bounds

We now show that the guarantees in Theorem 1 and Corollary 4 are tight up to numerical constants for adversarially constructed problems. We state the result for the cubic-regularization problem; corresponding lower bounds for the trust-region problem are immediate from the optimality gap relation (16).[1]

To state the result, we require a bit more notation. Let $\mathfrak{L}$ map cubic-regularization problem instances of the form $(A, b, \rho)$ to the quadruple $(\lambda_{\min}, \lambda_{\max}, \lambda_\star, \Delta) = \mathfrak{L}(A, b, \rho)$ such that $\lambda_{\min}, \lambda_{\max}$ are the extremal eigenvalues of $A$ and the solution $s_\star^{cr} = \operatorname{argmin}_x \hat{f}_{A,b,\rho}(x)$ satisfies $\rho\|s_\star^{cr}\| = \lambda_\star$, and $\hat{f}_{A,b,\rho}(0) - \hat{f}_{A,b,\rho}(s_\star^{cr}) = \Delta$. Similarly let $\mathfrak{L}'$ map an instance $(A, b, \rho)$ to the quadruple $(\lambda_{\min}, \lambda_{\max}, \tau, R)$ where now $\|s_\star^{cr}\| = R$ and $\|b\|/|u_{\min}^T b| = \tau$, with $u_{\min}$ an eigenvector of $A$ corresponding to eigenvalue $\lambda_{\min}$.

With this notation in hand, we state our lower bounds. (See supplemental section D for a proof.)

**Theorem 5.** *Let $d, t \in \mathbb{N}$ with $t < d$ and $\lambda_{\min}, \lambda_{\max}, \lambda_\star, \Delta$ be such that $\lambda_{\min} \leq \lambda_{\max}$, $\lambda_\star > (-\lambda_{\min})_+$, and $\Delta > 0$. There exists $(A, b, \rho)$ such that $\mathfrak{L}(A, b, \rho) = (\lambda_{\min}, \lambda_{\max}, \lambda_\star, \Delta)$ and for all $s \in \mathcal{K}_t(A, b)$,*

$$\hat{f}_{A,b,\rho}(s) - \hat{f}_{A,b,\rho}(s_\star^{cr}) > \frac{1}{K}\left[\hat{f}_{A,b,\rho}(0) - \hat{f}_{A,b,\rho}(s_\star^{cr})\right]\exp\left\{-\frac{4t}{\sqrt{\kappa}-1}\right\}, \qquad (19)$$

*where $K = 1 + \frac{\lambda_\star}{3(\lambda_\star + \lambda_{\min})}$ and $\kappa = \frac{\lambda_\star + \lambda_{\max}}{\lambda_\star + \lambda_{\min}}$. Alternatively, for any $\tau \geq 1$ and $R > 0$, there exists $(A, b, \rho)$ such that $\mathfrak{L}'(A, b, \rho) = (\lambda_{\min}, \lambda_{\max}, \tau, R)$ and for $s \in \mathcal{K}_t(A, b)$,*

$$\hat{f}_{A,b,\rho}(s) - \hat{f}_{A,b,\rho}(s_\star^{cr}) > \min\left\{(\lambda_{\max})_- - \lambda_{\min}, \frac{\lambda_{\max} - \lambda_{\min}}{16(t-\frac{1}{2})^2}\log^2\left(\frac{\|b\|^2}{(u_{\min}^T b)^2}\right)\right\}\frac{\|s_\star^{cr}\|^2}{32}, \quad (20)$$

*and*

$$\hat{f}_{A,b,\rho}(s) - \hat{f}_{A,b,\rho}(s_\star^{cr}) > \frac{(\lambda_{\max} - \lambda_{\min})\|s_\star^{cr}\|^2}{16(t+\frac{1}{2})^2}. \qquad (21)$$

The lower bounds (19) matches the linear convergence guarantee (17) to within a numerical constant, as we may choose $\lambda_{\max}, \lambda_{\min}$ and $\lambda_\star$ so that $\kappa$ is arbitrary and $K < 2$. Similarly, lower bounds (20) and (21) match the sublinear convergence rate (18) for $\lambda_{\min} < 0$ and $\lambda_{\min} \geq 0$ respectively. Our proof flows naturally from minimax characterizations of uniform polynomial approximations (Lemmas 4 and 5 in the supplement), which also play a crucial role in proving our upper bounds.

One consequence of the lower bound (19) is the existence of extremely badly conditioned instances, say with $\kappa = (100d)^2$ and $K = 3/2$, such that in the first $d-1$ iterations it is impossible to decrease the initial error by more than a factor of 2 (the initial error may be chosen arbitrarily large as well). However, since these instances have finite condition number we have $s_\star^{cr} \in \mathcal{K}_d(A, b)$, and so the error supposedly drops to 0 at the $d$th iteration. This seeming discontinuity stems from the fact that

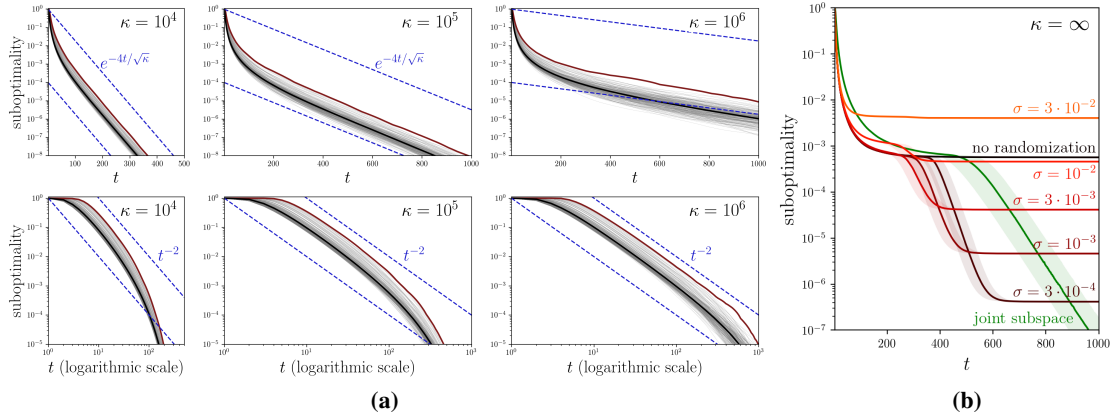

Figure 1: Optimality gap of Krylov subspace solutions on random cubic-regularization problems, versus subspace dimension $t$. **(a)** Columns show ensembles with different condition numbers $\kappa$, and rows differ by scaling of $t$. Thin lines indicate results for individual instances, and bold lines indicate ensemble median and maximum suboptimality. **(b)** Each line represents median suboptimality, and shaded regions represent inter-quartile range. Different lines correspond to different randomization settings.

in this case $s_\star^{\mathsf{cr}}$ depends on the Lanczos basis of $\mathcal{K}_d(A, b)$ through a very badly conditioned linear system and cannot be recovered with finite-precision arithmetic. Indeed, running Krylov subspace methods for $d$ iterations with inexact arithmetic often results in solutions that are very far from exact, while guarantees of the form (17) are more robust to roundoff errors [4, 13, 35].

While we state the lower bounds in Theorem 5 for points in the Krylov subspace $\mathcal{K}_t(A, b)$, a classical "resisting oracle" construction due to Nemirovski and Yudin [27, Chapter 7.2] (see also [26, §10.2.3]) shows that (for $d > 2t$) these lower bounds hold also for *any deterministic method* that accesses $A$ only through matrix-vector products, and computes a single matrix-vector product per iteration. The randomization we employ in Corollaries 2 and 3 breaks the lower bound (20) when $\lambda_{\min} < 0$ and $\|b\| / |u_{\min}^T b|$ is very large, so there is some substantial power from randomization in this case. However, Simchowitz [34] recently showed that randomization cannot break the lower bounds for convex quadratics ($\lambda_{\min} \geq 0$ and $\rho = 0$).

## 5  Numerical experiments

To see whether our analysis applies to non-worst case problem instances, we generate 5,000 random cubic-regularization problems with $d = 10^6$ and controlled condition number $\kappa = (\lambda_{\max} + \rho \|s_\star^{\mathsf{cr}}\|)/(\lambda_{\min} + \rho \|s_\star^{\mathsf{cr}}\|)$ (see Section E in the supplement for more details). We repeat the experiment three times with different values of $\kappa$ and summarize the results in Figure 1a. As seen in the figure, about 20 Lanczos iterations suffice to solve even the worst-conditioned instances to about 10% accuracy, and 100 iterations give accuracy better than 1%. Moreover, for $t \gtrsim \sqrt{\kappa}$, the approximation error decays exponentially with precisely the rate $4/\sqrt{\kappa}$ predicted by our analysis, for almost all the generated problems. For $t \ll \sqrt{\kappa}$, the error decays approximately as $t^{-2}$. We conclude that the rates characterized by Theorem 1 are relevant beyond the worst case.

We conduct an additional experiment to test the effect of randomization for "hard case" instances, where $\kappa = \infty$. We generate such problem instances (see details in Section E), and compare the joint subspace randomization scheme (Corollary 2) to the perturbation scheme (Corollary 3) with different perturbation magnitudes $\sigma$; the results are shown in Figure 1b. For any fixed target accuracy, some choices of $\sigma$ yield faster convergence than the joint subspace scheme. However, for any fixed $\sigma$ optimization eventually hits a noise floor due to the perturbation, while the joint subspace scheme continues to improve. Choosing $\sigma$ requires striking a balance: if too large the noise floor is high and might even be worse than no perturbation at all; if too small, escaping the unperturbed error level will take too long, and the method might falsely declare convergence. A practical heuristic for safely choosing $\sigma$ is an interesting topic for future research.

## Acknowledgments

We thank the anonymous reviewers for several helpful questions and suggestions. Both authors were supported by NSF-CAREER Award 1553086 and the Sloan Foundation. YC was partially supported by the Stanford Graduate Fellowship.

## Footnotes

[1]To obtain the correct prefactor in the trust-region equivalent of lower bound (19) we may use the fact that $\hat{f}_{A,b,\rho}(0) - \hat{f}_{A,b,\rho}(s_\star^{cr}) = \frac{1}{2}b^T A_{\lambda_\star}^{-1} b + \frac{\rho}{6}\|s_\star^{cr}\|^3 \geq \frac{1}{3}(\frac{1}{2}b^T A_{\lambda_\star}^{-1} b + \frac{\lambda_\star}{2}R^2) = \frac{1}{3}(f_{A,b}(0) - f_{A,b}(s_\star^{tr}))$.

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
