[Supplementary Material]

# Supplementary material

## A  Computing Krylov subspace solutions

Generic instances of the trust-region and cubic-regularized problems can be globally optimized by solving the one-dimensional equations

$$\left\|A_\lambda^{-1}b\right\| = R\,,\ \lambda > \max\{-\lambda_{\min}, 0\}. \tag{22}$$

and

$$\left\|A_\lambda^{-1}b\right\| = \lambda/\rho\,,\ \ \lambda \geq -\lambda_{\min}, \tag{23}$$

respectively. However, when $d$ is very large, even a single exact evaluation of $\left\|A_\lambda^{-1}b\right\|$ (which requires a direct linear system solution) can become prohibitively expensive.

In this case, a general approach to obtaining approximate solutions is to constrain the domain to a linear subspace $\mathcal{Q}_t \subset \mathbb{R}^d$ of dimension $t \ll d$. Let $Q_t \in \mathbb{R}^{d \times t}$ be an orthogonal basis for $\mathcal{Q}_t$ ($Q_t^T Q_t = I$). Finding the global minimizer in $\mathcal{Q}_t$ is equivalent to re-parameterizing $x$ as $x = Q_t \tilde{x}$ and solving for $\tilde{x} \in \mathbb{R}^t$, which is also equivalent to solving a $t$-dimensional problem instance with $\tilde{A} = Q_t^T A Q_t$ and $\tilde{b} = Q_t^T b$. For sufficiently large $d$, the time to solve such problems will be dominated by the $t$ matrix-vector products required to construct $\tilde{A}$.

In this paper we focus on the choice $\mathcal{Q}_t = \mathcal{K}_t(A, b)$ the Krylov subspace of order $t$. This choice offers a significant efficiency boost: we can efficiently construct a basis $Q_t$ for which $Q_t^T A Q_t$ is tridiagonal, using the Lanczos process, which consists of the following recursion, starting with $q_1 = b/\|b\|\,, q_0 = 0$,

$$\alpha_t = q_t^T A q_t\,,\ q_{t+1}' = A q_t - \alpha_t q_t - \beta_t q_{t-1}\,,\ \beta_{t+1} = \left\|q_{t+1}'\right\|\,,\ q_{t+1} = q_{t+1}'/\left\|q_{t+1}'\right\|.$$

The vectors $q_1, \ldots, q_t$ give the columns of $Q_t$ while $\alpha_1, \ldots, \alpha_t$ and $\beta_2, \ldots, \beta_t$ respectively give the diagonal and off-diagonal elements of the symmetric tridiagonal matrix $\tilde{A} = Q_t^T A Q_t$; this makes solving equations (22) and (23) easy. One straightforward approach is to directly compute the factorization $\tilde{A}$, which for a symmetric tridiagonal matrix of size $t$ takes $O(t \log t)$ time [10]. A more efficient approach—and the one used in practice—is to iteratively solve systems of the form $\tilde{A}_\lambda x = z$ and update $\lambda$ using Newton steps [11, 9]. Every tridiagonal system solution can be done in time $O(t)$, and the Newton steps are shown in [11, 9] to be globally linearly convergent, with local quadratic convergence. In our experience less than 20 Newton steps generally suffice to reach machine precision, and so the computational cost is essentially linear in $t$. It is also possible to avoid keeping $Q_t$ in memory (when $t \cdot d$ storage is too demanding) by running the Lanczos process twice, once for evaluating $\tilde{x}$ and again to obtain $x = Q_t \tilde{x}$.

The Lanczos process produces the same result as Gram-Schmidt orthonormalization of the vectors $\left[b, Ab, \ldots, A^{t-1}b\right]$ but uses the special structure of that matrix to avoid computing inner products that are known in advance to be zero. When run for many iterations, the Lanczos process has well-documented numerical stability issues [35]. However, in our setting we usually seek low to moderate accuracy solutions and will usually stop at $t < 100$, for which Lanczos is reasonably stable with floating point arithmetic even when $d$ is quite large. The application of the Lanczos process—which is typically used for eigenvector computation—in the context of regularized quadratic optimization is sometimes referred to as the generalized Lanczos process [17].

### A.1  Computing joint Krylov subspace solutions

To solve equations (22) and (23) in subspaces of the form

$$\mathcal{K}_{mt}(A, \{v_1, \ldots, v_m\}) := \mathrm{span}\{A^j v_i\}_{i \in \{1, \ldots, m\}, j \in \{0, \ldots, t-1\}}$$

we may use the block Lanczos method [12, 15], a natural generalization of the Lanczos method that creates an orthonormal basis for the subspace $\mathcal{K}_{mt}(A, \{v_1, \ldots, v_m\})$ in which $A$ has a block tridiagonal form. Overloading the notation defined above so that now $q_t \in \mathbb{R}^{d \times m}$ and $\alpha_t, \beta_t \in \mathbb{R}^{m \times m}$, the block Lanczos recursion is given by,

$$\alpha_t = q_t^T A q_t\,,\ q_{t+1}' = A q_t - q_t \alpha_t - q_{t-1}\beta_t^T\,,\ (q_{t+1}, \beta_{t+1}) = \mathrm{QR}(q_{t+1}').$$

where QR stands for the QR decomposition (i.e. if $(q, \beta) = \text{QR}(a)$ then $q$ is orthogonal, $\beta$ is upper diagonal and $a = q \cdot \beta$), and the initial conditions are that $q_1$ is an orthonormalized version of $[v_1, \ldots, v_m]$ and $q_0 = 0$. The matrix $\tilde{A} = Q_t^T A Q_t$ is now block tridiagonal, with the diagonal and sub-diagonal blocks given by $\{\alpha_i\}_{i \in \{1,\ldots,t\}}$ and $\{\beta_i\}_{i \in \{2,\ldots,t\}}$ respectively. Since the $\beta$ matrices are upper diagonal, $\tilde{A}$ is a symmetric banded matrix with $m$ non-zeros sub-diagonal bands. Such matrix admits fast Cholesky decomposition (in time linear in $m^2 t$), and consequently the Newton method described above is still efficient.

## B  Polynomial approximation results

In this section we state (and prove for ease of reference) two classical results on uniform polynomial approximation (cf. [24, 26]) that stand at the core of the technical development in this work.

**Lemma 4.** *Let $n \geq 1$ and $0 < \alpha \leq \beta$, and let $\kappa = \beta/\alpha$. Then*

$$\min_{p \in \mathcal{P}_n} \max_{x \in [\alpha,\beta]} |1 - xp(x)| = \mathfrak{T}_n(\kappa) := 2\left(\left(\frac{\sqrt{\kappa}+1}{\sqrt{\kappa}-1}\right)^n + \left(\frac{\sqrt{\kappa}-1}{\sqrt{\kappa}+1}\right)^n\right)^{-1}$$

*and*

$$2\left(e^{2n/(\sqrt{\kappa}-1)} + 1\right)^{-1} \leq \mathfrak{T}_n(\kappa) \leq 2e^{-2n/\sqrt{\kappa}}.$$

*Moreover, there exist $x_0, x_1, \ldots, x_n \in [\alpha, \beta]$ and probability distribution $\pi_0, \pi_1, \ldots \pi_n$ such that*

$$\min_{p \in \mathcal{P}_n} \sum_{k=0}^{n} \pi_k (1 - x_k p(x_k))^2 = [\mathfrak{T}_n(\kappa)]^2.$$

*Proof.* Let

$$T_n(x) = \begin{cases} \cos(n \arccos(x)) & |x| \leq 1 \\ \frac{1}{2}\left((x + \sqrt{x^2-1})^n + (x - \sqrt{x^2-1})^n\right) & |x| \geq 1 \end{cases}$$

denote the order $n$ Chebyshev polynomial of the first kind. We claim that $p^\star \in \mathcal{P}_n$ that solves the minimax problem $\min_{p \in \mathcal{P}_n} \max_{x \in [\alpha,\beta]} |1 - xp(x)|$ is given by

$$1 - xp^\star(x) = \mathfrak{T}_n(\kappa) \cdot T_n\left(\frac{\kappa + 1 - 2x/\alpha}{\kappa - 1}\right),$$

where $\mathfrak{T}_n(\kappa) = \left[T_n\left(\frac{\kappa+1}{\kappa-1}\right)\right]^{-1}$ guarantees that the RHS has value 1 at $x = 0$ and therefore $p^\star$ is well defined. Since clearly $|T_n(y)| \leq 1$ for every $y \in [-1, 1]$, we have that $\max_{x \in [\alpha,\beta]} |1 - xp^\star(x)| = \mathfrak{T}_n(\kappa)$.

We argue that $p^\star$ is optimal using the classical alternating signs argument, sometimes also referred to as Chebyshev's theorem. First, note that $T_n(y)$ has $n + 1$ extrema in $[-1, 1]$ (at $y_k = \cos(k\pi/n)$ for $k = 0, \ldots, n$) and that their values alternate between $-1$ and $1$ (i.e. $T_n(y_k) = (-1)^k$). Therefore, there exist $n + 1$ distinct points $x_0, x_1, \ldots, x_n \in [\alpha, \beta]$ for which $1 - x_i p^\star(x_k) = (-1)^k \mathfrak{T}_n(\kappa)$. Let $q \in \mathcal{P}_n$ satisfy $\max_{x \in [\alpha,\beta]} |1 - xq(x)| \leq \mathfrak{T}_n(\kappa)$. Then,

$$p^\star(x_k) - q(x_k) = \frac{[1 - x_k q(x_k)] - [1 - x_k p^\star(x_k)]}{x_k}$$

must be non-positive for even $k$ and non-negative for odd $k$, and therefore $p^\star - q$ must have at least $n$ roots in $[\alpha, \beta]$. However, $p^\star - q$ is a polynomial of degree at most $n - 1$ and can have $n$ roots only if it is identically 0, so we have that $q = p^\star$, proving that $p^\star$ is the unique solution of the minimax problem.

To see the upper and lower bounds on $\mathfrak{T}_n(\kappa)$, note that $\mathfrak{T}_n(\kappa) = 1/\cosh(n \log(1 + \frac{2}{\sqrt{\kappa}-1}))$, that $\frac{1}{2}e^{|y|} \leq \cosh(y) \leq \frac{1}{2}(e^{|y|} + 1)$, and that

$$\frac{2}{z} \leq \log\left(1 + \frac{2}{z-1}\right) \leq \frac{2}{z-1}$$

for all $z > 1$, where the lower bound above can seen by comparing derivatives.

To see the final part of the lemma, let $x_0, x_1, \ldots, x_n \in [\alpha, \beta]$ be the points constructed in the optimality argument above, and note that this argument continues to hold if the inner maximization is restricted to these points. Therefore,

$$\min_{p \in \mathcal{P}_n} \max_{0 \le k \le n} (1 - x_k p(x_k))^2 = \left[ \min_{p \in \mathcal{P}_n} \max_{0 \le k \le n} |1 - x_k p(x_k)| \right]^2 = [\mathfrak{T}_n(\kappa)]^2.$$

Letting $\Delta_{n+1}$ denote the probability simplex with $n+1$ variables, we may write

$$\max_{0 \le k \le n} (1 - x_k p(x_k))^2 = \max_{\mu \in \Delta_{n+1}} \sum_{k=0}^{n} \mu_k (1 - x_k p(x_k))^2.$$

Finally, noting that the objective $\sum_{k=0}^{n} \mu_k (1 - x_k p(x_k))^2$ is linear (and hence concave) in $\mu$ and convex in (the coefficients of) $p$, we may use Von-Neumann's lemma and swap the min and max above, writing

$$\max_{\mu \in \Delta_{n+1}} \min_{p \in \mathcal{P}_n} \sum_{k=0}^{n} \mu_k (1 - x_k p(x_k))^2 = \min_{p \in \mathcal{P}_n} \max_{\mu \in \Delta_{n+1}} \sum_{k=0}^{n} \mu_k (1 - x_k p(x_k))^2 = [\mathfrak{T}_n(\kappa)]^2.$$

Letting $\pi$ denote the distribution attaining the outer maximum, we get the desired result. We remark in passing that $\pi$ may be constructed explicitly using the orthogonality principle of least squares estimation and orthogonality relations of Chebyshev polynomials. $\qquad\square$

**Lemma 5.** *Let $n \ge 1$ and $0 < \alpha \le \beta$, let $\kappa = \beta/\alpha$ and define $w(x) := \sqrt{x - \alpha}$. Then*

$$\min_{p \in \mathcal{P}_n} \max_{x \in [\alpha, \beta]} w(x) |1 - x p(x)| = \mathfrak{U}_n(\kappa) := 2\sqrt{\alpha} \left( \left( \frac{\sqrt{\kappa} + 1}{\sqrt{\kappa} - 1} \right)^{n + \frac{1}{2}} - \left( \frac{\sqrt{\kappa} - 1}{\sqrt{\kappa} + 1} \right)^{n + \frac{1}{2}} \right)^{-1}$$

*and*

$$2\sqrt{\alpha} \left( e^{2(2n+1)/(\sqrt{\kappa} - 1)} - 1 \right)^{-\frac{1}{2}} \le \mathfrak{U}_n(\kappa) \le 2\sqrt{\alpha} \left( e^{2(2n+1)/\sqrt{\kappa}} - 2 \right)^{-\frac{1}{2}}.$$

*Moreover, there exist $x_0, x_1, \ldots, x_n \in [\alpha, \beta]$ and probability distribution $\pi_0, \pi_1, \ldots \pi_n$ such that*

$$\min_{p \in \mathcal{P}_n} \sum_{k=0}^{n} \pi_k w^2(x_k) (1 - x_k p(x_k))^2 = [\mathfrak{U}_n(\kappa)]^2.$$

*Proof.* Let

$$U_n(x) = \begin{cases} \frac{1}{\sqrt{1 - x^2}} \sin((n+1) \arccos(x)) & |x| \le 1 \\ \frac{1}{2\sqrt{x^2 - 1}} \left( (x + \sqrt{x^2 - 1})^{n+1} - (x - \sqrt{x^2 - 1})^{n+1} \right) & |x| \ge 1 \end{cases}$$

denote the order $n$ Chebyshev polynomial of the second kind. We claim that $p^\star \in \mathcal{P}_n$ that solves the minimax problem $\min_{p \in \mathcal{P}_n} \max_{x \in [\alpha, \beta]} (x - \alpha)^{1/2} |1 - x p(x)|$ is given by

$$1 - x p^\star(x) = \frac{\mathfrak{U}_n(\kappa)}{w(\beta)} \cdot U_{2n} \left( \sqrt{\frac{\kappa - x/\alpha}{\kappa - 1}} \right),$$

where $\mathfrak{U}_n(\kappa) = w(\beta) \left[ U_{2n} \left( \sqrt{\frac{\kappa}{\kappa - 1}} \right) \right]^{-1}$ guarantees that the RHS has value 1 at $x = 0$ and therefore $p^\star$ is well defined (note that $U_{2n}(\cdot)$ is an even polynomial and therefore $U_{2n}(\sqrt{\cdot})$ is a polynomial of degree $n$). For $x \in [\alpha, \beta]$, we have by the definition of $p^*$ and the expression for $U_{2n}$,

$$w(x)(1 - x p^\star(x)) = \mathfrak{U}_n(\kappa) \cdot \sin \left( (2n+1) \arccos \left( \sqrt{\frac{\kappa - x/\alpha}{\kappa - 1}} \right) \right).$$

Therefore, we have that $w(x) |1 - x p^\star(x)| \le \mathfrak{U}_n(\kappa)$ for every $x \in [\alpha, \beta]$, and moreover we have that $w(x_k)(1 - x_k p^\star(x_k)) = (-1)^k \cdot \mathfrak{U}_n(\kappa)$, for the points $x_0, \ldots x_n \in [\alpha, \beta]$ satisfying

$$\sqrt{\frac{\kappa - x_k/\alpha}{\kappa - 1}} = \cos \left( \frac{\pi}{2} \cdot \frac{2k + 1}{2n + 1} \right).$$

Hence, the alternating signs argument from the proof of Lemma 4 holds here as well and we have that $p^\star$ is optimal and that $\min_{p \in \mathcal{P}_n} \max_{x \in [\alpha, \beta]} w(x)|1 - xp(x)| = \mathfrak{U}_n(\kappa)$.

To see the upper and lower bounds on $\mathfrak{U}_n(\kappa)$, note that $\mathfrak{U}_n(\kappa) = \sqrt{\alpha} / \sinh((n + \frac{1}{2}) \log(1 + \frac{2}{\sqrt{\kappa}-1}))$, that for $y \geq 0$, $\sinh(y) = \frac{1}{\sqrt{2}} \sqrt{\cosh(2y) - 1}$ gives $\frac{1}{2} \sqrt{e^{2y} - 2} \leq \sinh(y) \leq \frac{1}{2} \sqrt{e^{2y} - 1}$, and that (as in Lemma 4) $\frac{2}{z} \leq \log\left(1 + \frac{2}{z-1}\right) \leq \frac{2}{z-1}$.

The final part of the lemma follows exactly as in Lemma 4. $\qquad\square$

## C  Proofs from Section 2

### C.1  Proof of auxiliary lemmas

**Lemma 1** (Approximate matrix inverse). *Let* $\alpha, \beta$ *satisfy* $0 < \alpha \leq \beta$, *and let* $\kappa = \beta/\alpha$. *For any* $t \geq 1$ *there exists a polynomial* $p$ *of degree at most* $t - 1$, *such that for every* $M$ *satisfying* $\alpha I \preceq M \preceq \beta I$,

$$\|I - Mp(M)\| \leq 2e^{-2t/\sqrt{\kappa}}.$$

*Proof.* This is an immediate consequence of Lemma 4, as

$$\min_{p \in \mathcal{P}_t} \max_{\alpha I \preceq M \preceq \beta I} \|I - Mp(M)\| = \min_{p \in \mathcal{P}_t} \max_{\lambda \in [\alpha, \beta]} |1 - \lambda \cdot p(\lambda)| = \mathfrak{T}_t(\kappa).$$

$\qquad\square$

**Lemma 2** (Convex trust-region problem). *Let* $t \geq 1$, $M \succeq 0$, $v \in \mathbb{R}^d$ *and* $r \geq 0$, *and let* $f_{M,v}(x) = \frac{1}{2} x^T M x + v^T x$. *There exists* $x_t \in \mathcal{K}_t(M, v)$ *such that*

$$\|x_t\| \leq r \quad \text{and} \quad f_{M,v}(x_t) - \min_{\|x\| \leq r} f_{M,v}(x) \leq \frac{4\lambda_{\max}(M) \cdot r^2}{(t+1)^2}.$$

*Proof.* Let $g : \mathbb{R}^d \to \mathbb{R}$ be convex with $L$-Lipschitz gradient and let $Q \subseteq \mathbb{R}^d$ be a convex set containing the point 0. Consider Nesterov's accelerated gradient method for minimization of $g$, which comprises the following recursion [28, Scheme (2.2.17)],

$$x_{k+1} = \min_{x \in Q} \left\{ x^T \nabla g(y_k) + \frac{L}{2} \|x - y_k\|^2 \right\} = \Pi_Q \left( y_k - \frac{1}{L} \nabla g(y_k) \right)$$

$$\alpha_{k+1}^2 / (1 - \alpha_{k+1}) = \alpha_k^2 \Rightarrow \alpha_{k+1} = -\frac{\alpha_k^2}{2} + \frac{\alpha_k^2}{2} \sqrt{1 + \frac{4}{\alpha_k^2}}$$

$$y_{k+1} = x_{k+1} + \alpha_{k+1}(\alpha_k^{-1} - 1)(x_{k+1} - x_k),$$

where $\Pi_Q(\cdot)$ is the Euclidean projection to $Q$. Letting $\alpha_0 = 1$ and $y_0 = x_0 = 0$, and letting $x^\star$ denote any minimizer of $g$ in $Q$, the analysis of Tseng [37, Corollary 2(b)] gives[2],

$$g(x_t) - g(x^\star) \leq \frac{4L \max_{z \in Q} \|z\|^2}{(t+1)^2}. \tag{24}$$

Taking $g = f_{M,v}$ and $Q = B_r = \{x \mid \|x\| \leq r\}$, we note that $f_{M,v}$ is convex with $L := \lambda_{\max}(M)$-Lipschitz gradient, and that the projection step guarantees that $\|x_t\| \leq r$ for every $t$. Therefore, to establish the lemma it remains only to argue that $x_t$ as defined above is in $\mathcal{K}_t(M, v)$; we shall see this by simple induction, whose basis is $y_0, x_0 \in \mathcal{K}_0(M, v) = \{0\}$. Assume now that $y_k, x_k \in \mathcal{K}_k(M, v)$ for some $k \geq 0$. This implies

$$y_k - \frac{1}{L} \nabla g(y_k) = y_k - \frac{1}{L} A y_k - \frac{1}{L} v \in \mathcal{K}_{k+1}(M, v).$$

Further, note that projection to the Euclidean ball $B_r$ is simply scaling:

$$\Pi_Q(z) = \Pi_{B_r}(z) = \frac{r}{\max\{r, \|z\|\}} \cdot z,$$

and therefore $x_{k+1} \in \mathcal{K}_{k+1}(M, v)$. Finally, $y_{k+1}$ is simply a linear combination of $x_{k+1}$ and $x_k$ and therefore is also in $\mathcal{K}_{k+1}(M, v)$, concluding the induction and the proof. $\qquad\square$

A bound similar to (24) appears in Nesterov's earlier analysis [28, Theorem 2.2.3], but with an the additional factor proportional to $g(0) - g(x^\star)$ which is not immediately upper bounded by $\frac{1}{2}L \max_{z \in Q} \|z\|^2$ due to the constraint $z \in Q$. The bound (24) also appears in later work of Allen-Zhu and Orecchia [2].

**Lemma 3** (Finding eigenvectors, [24, Theorem 4.2]). *Let $M \succeq 0$ be such that $u^T M u = 0$ for some unit vector $u \in \mathbb{R}^d$, and let $v \in \mathbb{R}^d$. For every $t \geq 1$ there exists $z_t \in \mathcal{K}_t(M, v)$ such that*

$$\|z_t\| = 1 \ \text{ and } \ z_t^T M z_t \leq \frac{\|M\|}{16(t - \frac{1}{2})^2} \log^2\left(-2 + 4\frac{\|v\|^2}{(u^T v)^2}\right).$$

*Proof.* Let $\lambda_{(1)} \leq \lambda_{(2)} \leq \cdots \leq \lambda_{(d)}$ denote the eigenvalues of $M$ and let $u_1, u_2, \ldots, u_d$ denote their corresponding (orthonormal) eigenvectors. By our assumption $\lambda_{(1)} = 0$ and we have also $\lambda_{(d)} = \|M\|$. We let

$$v_{(i)} := u_i^T v$$

denote the component of $v$ in the eigenbasis of $M$. Define

$$\mathsf{err}_t := \min_{p \in \mathcal{P}_t} \frac{(p(M)v)^T M p(M)v}{\|p(M)v\|^2} = \min_{p \in \mathcal{P}_t} \frac{\sum_{i=1}^d v_{(i)}^2 p^2(\lambda_{(i)})\lambda_{(i)}}{\sum_{i=1}^d v_{(i)}^2 p^2(\lambda_{(i)})},$$

and let $q \in \mathcal{P}_t$ attain the minimum above. Setting $z_t = q(M)v / \|q(M)v\|$, we see that

$$\mathsf{err}_t = z_t^T M z_t = \frac{\sum_{i=1}^d v_{(i)}^2 q^2(\lambda_{(i)})\lambda_{(i)}}{\sum_{i=1}^d v_{(i)}^2 q^2(\lambda_{(i)})},$$

and so our proof comprises of bounding $\mathsf{err}_t$ from above.

We invoke Lemma 5 with $n = t - 1$, $\alpha = \mathsf{err}_t$ and $\beta = \lambda_{(d)} = \|M\|$; let $\tilde{q}(x) = 1 - xp^\star(x) \in \mathcal{P}_t$ be the polynomial for which the Lemma guarantees

$$\max_{x \in [\mathsf{err}_t, \lambda_{(d)}]} (x - \mathsf{err}_t)^{1/2} |\tilde{q}(x)| = \mathfrak{U}_{t-1}(\kappa).$$

By the optimality of $q$, we have that

$$\mathsf{err}_t \leq \frac{\sum_{i=1}^d v_{(i)}^2 \tilde{q}^2(\lambda_{(i)})\lambda_{(i)}}{\sum_{i=1}^d v_{(i)}^2 \tilde{q}^2(\lambda_{(i)})}.$$

Rearranging and noting that $\tilde{q}(\lambda_{(1)}) = \tilde{q}(0) = 1$, we obtain

$$\mathsf{err}_t \leq \sum_{i=2}^d \frac{v_{(i)}^2}{v_{(1)}^2}(\lambda_{(i)} - \mathsf{err}_t)\tilde{q}^2(\lambda_{(i)}) \leq \frac{\|v\|^2 - v_{(1)}^2}{v_{(1)}^2} \max_{\lambda \in [\mathsf{err}_t, \lambda_{(d)}]} (\lambda - \mathsf{err}_t)\tilde{q}^2(\lambda) = \left(\frac{\|v\|^2}{v_{(1)}^2} - 1\right)[\mathfrak{U}_{t-1}(\kappa)]^2.$$

Lemma 5 provides the bound

$$[\mathfrak{U}_{t-1}(\kappa)]^2 \leq \frac{4\mathsf{err}_t}{e^{2(2t-1)\sqrt{\mathsf{err}_t/\|M\|}} - 2}.$$

Substituting the upper bound into $\mathsf{err}_t \leq \left(\frac{\|v\|^2}{v_{(1)}^2} - 1\right)[\mathfrak{U}_{t-1}(\kappa)]^2$ and rearranging gives the result. $\qquad\square$

## C.2 Proof of Corollary 3

**Corollary 3.** *Let $v$ be uniformly distributed on the unit sphere in $\mathbb{R}^d$, let $\sigma > 0$ and let*
$$\tilde{b} = b + \sigma \cdot v.$$
*Let $\tilde{s}_t^{\mathrm{tr}} \in \mathrm{argmin}_{x \in \mathcal{K}_t(A, \tilde{b}), \|x\| \leq R} f_{A, \tilde{b}}(x) := \frac{1}{2} x^T A x + \tilde{b}^T x$. For any $\delta > 0$,*

$$f_{A,b}(\tilde{s}_t^{\mathrm{tr}}) - f_{A,b}(s_\star^{\mathrm{tr}}) \leq \frac{(\lambda_{\max} - \lambda_{\min})R^2}{(t - \frac{1}{2})^2} \left[ 4 + \frac{\mathbb{I}_{\{\lambda_{\min} < 0\}}}{2} \log^2 \left( \frac{2\|\tilde{b}\| \sqrt{d}}{\sigma \delta} \right) \right] + 2\sigma R \quad (14)$$

*with probability at least $1 - \delta$ with respect to the random choice of $v$.*

*Proof.* Let $\tilde{x}_{\mathrm{tr}}^\star \in \mathrm{argmin}_{x \in \mathcal{K}_t(A, \tilde{b}), \|x\| \leq R} f_{A, \tilde{b}}(x)$ be a solution to the perturbed problem. Since $v$ is a unit vector, for any feasible $x$ we have

$$f_{A,b}(x) - f_{A,b}(s_\star^{\mathrm{tr}}) = f_{A,\tilde{b}}(x) - f_{A,\tilde{b}}(s_\star^{\mathrm{tr}}) + \sigma \cdot v^T(s_\star^{\mathrm{tr}} - x) \leq f_{A,\tilde{b}}(x) - f_{A,\tilde{b}}(s_\star^{\mathrm{tr}}) + 2\sigma R$$
$$\leq f_{A,\tilde{b}}(x) - f_{A,\tilde{b}}(\tilde{x}_{\mathrm{tr}}^\star) + 2\sigma R, \quad (25)$$

and so it suffices to argue about the perturbed optimality gap $f_{A,\tilde{b}}(\tilde{s}_t^{\mathrm{tr}}) - f_{A,\tilde{b}}(s_\star^{\mathrm{tr}})$.

Applying the bound (5) on the perturbed problem gives us

$$f_{A,\tilde{b}}(\tilde{s}_t^{\mathrm{tr}}) - f_{A,\tilde{b}}(\tilde{x}_{\mathrm{tr}}^\star) \leq \frac{(\lambda_{\max} - \lambda_{\min})R^2}{(t - \frac{1}{2})^2} \left[ 4 + \frac{\mathbb{I}_{\{\lambda_{\min} < 0\}}}{2} \log^2 \left( 2 \frac{\|\tilde{b}\|}{|u_{\min}^T \tilde{b}|} \right) \right], \quad (26)$$

and a simple argument on the density of $u_{\min}^T \tilde{b}$ (cf. [5, Lemma 4.6]) shows that

$$|u_{\min}^T \tilde{b}| \geq \frac{\sigma \cdot \delta}{\sqrt{d}} \quad \text{with probability at least } 1 - \delta. \quad (27)$$

Combining the bounds (25), (26) and (27) gives the result (14). $\square$

# D    Proof of lower bounds

In what follows, we break Theorem 5 into two parts, one for the linear convergence lower bound (19) and one for the sublinear lower bounds (20) and (21). We restate each sub-theorem in a way that clearly shows our control over problem-dependent parameters when constructing the hard problem instances. In our proofs we will make use of the following expression for the optimality gap in the cubic-regularization problem,

$$\hat{f}_{A,b,\rho}(x) - \hat{f}_{A,b,\rho}(s_\star^{\mathrm{cr}}) = \frac{1}{2}(x - s_\star^{\mathrm{cr}})^T A_{\rho \|s_\star^{\mathrm{cr}}\|}(x - s_\star^{\mathrm{cr}}) + \frac{\rho}{6} (\|s_\star^{\mathrm{cr}}\| - \|x\|)^2 (\|s_\star^{\mathrm{cr}}\| + 2\|x\|), \quad (28)$$

where $A_{\rho \|s_\star^{\mathrm{cr}}\|} = A + \rho \|s_\star^{\mathrm{cr}}\| I$.

## D.1    Proof of linear convergence lower bound

**Theorem 5, part I.** *Let $\lambda_{\min}, \lambda_{\max}, \lambda_\star, \Delta \in \mathbb{R}$ such that $\lambda_{\min} \leq \lambda_{\max}$, $\lambda_\star > \max\{-\min, 0\}$ and $R, \Delta > 0$. For every $t \geq 1$ and every $d > t$ there exists $A \in \mathbb{R}^{d \times d}$, $b \in \mathbb{R}^d$ and $\rho > 0$ such that*

- *all eigenvalues of $A$ are in $[\lambda_{\min}, \lambda_{\max}]$,*

- *the solution $s_\star^{\mathrm{cr}} = \mathrm{argmin}_{x \in \mathbb{R}^d} \hat{f}_{A,b,\rho}(x)$ satisfies $\rho \|s_\star^{\mathrm{cr}}\| = \lambda_\star$,*

- *$\hat{f}_{A,b,\rho}(0) - \hat{f}_{A,b,\rho}(s_\star^{\mathrm{cr}}) = \Delta$, and*

$$\hat{f}_{A,b,\rho}(s) - \hat{f}_{A,b,\rho}(s_\star^{\mathrm{cr}}) > \left( 1 + \frac{\rho \|s_\star^{\mathrm{cr}}\|}{3(\rho \|s_\star^{\mathrm{cr}}\| + \lambda_{\min})} \right)^{-1} \left[ \hat{f}_{A,b,\rho}(0) - \hat{f}_{A,b,\rho}(s_\star^{\mathrm{cr}}) \right] \exp \left\{ - \frac{4t}{\sqrt{\frac{\rho \|s_\star^{\mathrm{cr}}\| + \lambda_{\max}}{\rho \|s_\star^{\mathrm{cr}}\| + \lambda_{\min}}} - 1} \right\}.$$

*for every $s \in \mathcal{K}_t(A, b)$.*

*Proof.* From Lemma 4 with $\alpha = \lambda_\star + \lambda_{\min}$, $\beta = \lambda_\star + \lambda_{\max}$ and $n = t$, we have that there exist $\xi_0, \dots, \xi_t \in [\alpha, \beta]$ and probability distribution $\pi_0, \dots, \pi_t$ such that

$$\min_{p \in \mathcal{P}_t} \sum_{k=0}^{t} \pi_k (1 - \xi_k p(\xi_k))^2 \geq e^{-4t/(\sqrt{\kappa}-1)},$$

where $\kappa = \beta/\alpha = (\lambda_{\max} + \lambda_\star)/(\lambda_{\min} + \lambda_\star)$. We let $\xi$ and $\sqrt{\pi}$ denote vectors with entries $\xi_0, \dots, \xi_t$ and $\sqrt{\pi_0}, \dots, \sqrt{\pi_t}$ respectively.

To construct the problem instance $(A, b, \rho)$ we assume without loss of generality $d = t+1$, as higher dimensional instances can be obtained by zero-padding a $(t+1)$-dimensional construction. We set

$$A = \operatorname{diag}(\xi - \lambda_\star), \quad b = \mu A_{\lambda_\star}^{1/2} \sqrt{\pi} \text{ and } \rho = \lambda_\star / \left\| A_{\lambda_\star}^{-1} b \right\|,$$

where we will choose $\mu > 0$ to set the value of $\hat{f}_{A,b,\rho}(0) - \hat{f}_{A,b,\rho}(s_\star^{\mathrm{cr}})$. First, we note that for any value of $\mu$ our choice of $\rho$ guarantees that $\left\| A_{\lambda_\star}^{-1} b \right\| = \lambda_\star / \rho$, making $s_\star^{\mathrm{cr}} = -A_{\lambda_\star}^{-1} b$ the unique global minimizer of $\hat{f}_{A,b,\rho}$. We therefore have by equation (28)

$$\hat{f}_{A,b,\rho}(0) - \hat{f}_{A,b,\rho}(s_\star^{\mathrm{cr}}) = \frac{1}{2} s_\star^{\mathrm{cr}\,T} A_{\lambda_\star} s_\star^{\mathrm{cr}} + \frac{\rho \left\| s_\star^{\mathrm{cr}} \right\|}{6} \| s_\star^{\mathrm{cr}} \|^2 = \frac{\mu^2}{2} \left( 1 + \frac{\lambda_\star}{3} \sqrt{\pi}^T A_{\lambda_\star}^{-1} \sqrt{\pi} \right),$$

so for every $\Delta > 0$ there is $\mu$ for which $\hat{f}_{A,b,\rho}(0) - \hat{f}_{A,b,\rho}(s_\star^{\mathrm{cr}}) = \Delta$. Noting that $\sqrt{\pi}^T A_{\lambda_\star}^{-1} \sqrt{\pi} \leq (\lambda_\star + \lambda_{\min})^{-1} \| \sqrt{\pi} \|^2 = (\lambda_\star + \lambda_{\min})^{-1}$, we also have

$$\frac{\mu^2}{2} \geq \Delta \left( 1 + \frac{\lambda_\star}{3(\lambda_\star + \lambda_{\min})} \right)^{-1}.$$

Now, every $s \in \mathcal{K}_t(A, b)$ is of the form $s = -p(A_{\lambda_\star})b$ for $p \in \mathcal{P}_t$, and using equation (28) again we have

$$\hat{f}_{A,b,\rho}(s) - \hat{f}_{A,b,\rho}(s_\star^{\mathrm{cr}}) \geq \frac{1}{2} \left\| A_{\lambda_\star}^{1/2}(s - s_\star^{\mathrm{cr}}) \right\|^2 \overset{(a)}{=} \frac{1}{2} \left\| (I - A_{\lambda_\star} p(A_{\lambda_\star})) A_{\lambda_\star}^{-1/2} b \right\|^2$$

$$\overset{(b)}{=} \frac{\mu^2}{2} \sum_{k=0}^{n} \pi_k (1 - \xi_k p(\xi_k))^2 \overset{(c)}{\geq} \frac{\mu^2}{2} e^{-4t/(\sqrt{\kappa}-1)},$$

where in $(a)$ we substituted $s = -p(A_{\lambda_\star})b$ and $s_\star^{\mathrm{cr}} = -A_{\lambda_\star}^{-1}b$, in $(b)$ we used our construction of $A$ and $b$, and in $(c)$ we used the guarantee from Lemma 4. The result follows from substituting our lower bound on $\mu^2$ and recalling that $\lambda_\star = \rho \| s_\star^{\mathrm{cr}} \|$. $\qquad\square$

### D.2 A lower bound for finding eigenvectors

The "non-convex" lower bound is in its heart a statement about the difficulty of approximating an extremal eigenvector in a Krylov subspace, which we state explicitly here. The proof of the lemma consists of applying "in reverse" the same polynomial approximation result (Lemma 5) that Kuczynski and Wozniakowski [24] use for proving upper bounds on finding eigenvector with the Lanczos method (which we state as Lemma 3).

**Lemma 6** (Finding eigenvectors: lower bound). *For every $d > 0$, vector $v \in \mathbb{R}^d$, unit vector $u \in \mathbb{R}^d$ and $t < d$, there exists matrix $M \in \mathbb{R}^{d \times d}$ such that $M \succeq 0$, $Mu = 0$, and for every $z \in \mathcal{K}_t(M, v)$,*

$$\frac{z_t^T M z_t}{\|M\| \|z_t\|^2} \geq \min \left\{ \frac{1}{4}, \frac{1}{64(t - \frac{1}{2})^2} \log^2 \left( -3 + 4 \frac{\|v\|^2}{(u^T v)^2} \right) \right\}.$$

*Proof.* We take $\|M\| = 1$ without loss of generality; results for arbitrary norms of $M$ follow by scaling the construction below. Define

$$\mathrm{err}_t := \min \left\{ \frac{1}{4}, \frac{1}{64(t - \frac{1}{2})^2} \log^2 \left( -3 + 4 \frac{\|v\|^2}{(u^T v)^2} \right) \right\}. \tag{29}$$

We apply Lemma 5 with $n = t - 1$, $\alpha = \mathsf{err}_t$ and $\beta = 1$, to obtain $\xi_1, \ldots, \xi_t \in [\mathsf{err}_t, 1]$ and probability distribution $\pi_1, \ldots, \pi_t$ such that

$$\min_{p \in \mathcal{P}_{t-1}} \sum_{k=1}^{t} \pi_k (\xi_k - \mathsf{err}_t)(1 - \xi_k p(\xi_k))^2 \geq \frac{4\mathsf{err}_t}{e^{2(2t-1)/(\frac{1}{\sqrt{\mathsf{err}_t}} - 1)} - 1}. \tag{30}$$

We assume without loss of generality that $d = t + 1$ (otherwise we zero-pad), and construct $M$ as follows. First, we take the eigenvalues of $M$ to be $0, \xi_1, \ldots, \xi_t$, satisfying $0 \preceq M \preceq I$. Next, we let $u$ be the eigenvector of $M$ corresponding to eigenvalue 0, satisfying $Mu = 0$. Finally, for $i = 1, \ldots, t$ we choose the eigenvector $u_i$ corresponding to eigenvalue $\xi_i$ such that $(u_i^T v)^2 = \pi_i(\|v\|^2 - (u^T v)^2)$.

Assume by contradiction

$$\min_{z \in \mathcal{K}_t(M,v)} \frac{z^T M z}{\|z\|^2} < \mathsf{err}_t, \tag{31}$$

and let $q \in \mathcal{P}_t$ be be such that

$$\frac{\sum_{i=1}^{t} \xi_i q^2(\xi_i)(u_i^T v)^2}{q^2(0)(u^T v)^2 + \sum_{i=1}^{t} q^2(\xi_i)(u_i^T v)^2} = \frac{(q(M)v)^T M q(M)v}{\|q(M)v\|^2} = \min_{z \in \mathcal{K}_t(M,v)} \frac{z^T M z}{\|z\|^2} < \mathsf{err}_t.$$

Rearranging, using $(u_i^T v)^2 = \pi_i(\|v\|^2 - (u^T v)^2)$, and letting $\tilde{q}(x) = q(x)/q(0)$, we have that

$$\mathsf{err}_t > \left( \frac{\|v\|^2}{(u^T v)^2} - 1 \right) \sum_{i=1}^{t} \pi_i(\xi_i - \mathsf{err}_t)\tilde{q}^2(\xi_i) \geq \left( \frac{\|v\|^2}{(u^T v)^2} - 1 \right) \frac{4\mathsf{err}_t}{e^{2(2t-1)/(\frac{1}{\sqrt{\mathsf{err}_t}} - 1)} - 1}.$$

where in the last transition we used that $\tilde{q}(0) = 1$ and therefore it is of the form $1 - xp(x)$ for some $p \in \mathcal{P}_{t-1}$, so the lower bound (30) applies. Rearranging gives

$$\mathsf{err}_t > h \left( \frac{1}{16(t - \frac{1}{2})^2} \log^2 \left( -3 + 4\frac{\|v\|^2}{(u^T v)^2} \right) \right), \quad h(x) = \frac{x}{(1 + \sqrt{x})^2}.$$

Using $h(x) \geq \frac{1}{4}\min\{1, x\}$ and the definition (29) of $\mathsf{err}_t$, we see that the above bound gives the contradiction $\mathsf{err}_t > \mathsf{err}_t$ and therefore assumption (31) must be false and we have the desired result $\min_{z \in \mathcal{K}_t(M,v)} \frac{z^T M z}{\|z\|^2} \geq \mathsf{err}_t$. $\square$

### D.3 Proof of sublinear convergence lower bound

**Theorem 5, part II.** *Let $\lambda_{\min}, \lambda_{\max}, R, \tau \in \mathbb{R}$ such that $\lambda_{\min} \leq \lambda_{\max}$, $\tau \geq 1$ and $R > 0$. For every $t \geq 1$ and every $d > t$ there exists $A \in \mathbb{R}^{d \times d}$, $b \in \mathbb{R}^d$ and $\rho > 0$ such that*

- *all eigenvalues of $A$ are in $[\lambda_{\min}, \lambda_{\max}]$,*

- *the solution $s_\star^{\mathsf{cr}} = \mathrm{argmin}_{x \in \mathbb{R}^d} \hat{f}_{A,b,\rho}(x)$ satisfies $\|s_\star^{\mathsf{cr}}\| = R$,*

- *there exists unit eigenvector $u_{\min}$ such that $u_{\min}^T A u_{\min} = \lambda_{\min}$ and $\frac{\|b\|}{|u_{\min}^T b|} = \tau$, and*

$$\hat{f}_{A,b,\rho}(s) - \hat{f}_{A,b,\rho}(s_\star^{\mathsf{cr}}) > \min\left\{ \lambda_{\max}^- - \lambda_{\min} , \frac{\lambda_{\max} - \lambda_{\min}}{16(t - \frac{1}{2})^2} \log^2 \left( \frac{\|b\|^2}{(u_{\min}^T b)^2} \right) \right\} \frac{\|s_\star^{\mathsf{cr}}\|^2}{32},$$

*where $\lambda_{\max}^- = \min\{\lambda_{\max}, 0\}$, and*

$$\hat{f}_{A,b,\rho}(s) - \hat{f}_{A,b,\rho}(s_\star^{\mathsf{cr}}) > \frac{(\lambda_{\max} - \lambda_{\min})\|s_\star^{\mathsf{cr}}\|^2}{16(t + \frac{1}{2})^2}.$$

*for every $s \in \mathcal{K}_t(A, b)$.*

*Proof.* We begin with the first, "non-convex" bound, which is essentially a reduction to the eigenvector problem. Here we assume $\lambda_{\min} \leq 0$ as otherwise the lower bound is vacuous. We use Lemma 6 to construct $M \in \mathbb{R}^{d \times d}$ and unit vectors $u_{\min}, v \in R^d$ such that $M \succeq 0$, $\|M\| = \lambda_{\max} - \lambda_{\min}$, $Mu_{\min} = 0$, $\|v\| / |u_{\min}^T v| = \tau$ and for every $z \in \mathcal{K}_t(M, v)$

$$\frac{z^T M z}{\|z\|^2} \geq \frac{\lambda_{\max} - \lambda_{\min}}{4} \min\left\{ 1, \frac{1}{16(t - \frac{1}{2})^2} \log^2\left( -3 + 4\frac{\|v\|^2}{(u_{\min}^T v)^2} \right) \right\}$$

$$\geq \frac{1}{4} \min\left\{ \lambda_{\max}^- - \lambda_{\min}, \frac{\lambda_{\max} - \lambda_{\min}}{16(t - \frac{1}{2})^2} \log^2\left( \frac{\|v\|^2}{(u_{\min}^T v)^2} \right) \right\} := \epsilon_t, \tag{32}$$

where $\lambda_{\max}^- = \min\{\lambda_{\max}, 0\}$. We let $\varepsilon > 0$ be a parameter to be specified later. We let

$$\lambda_\star = -\lambda_{\min} + \varepsilon$$

and construct the cubic regularization instance as follows

$$A = M + \lambda_{\min} I , \ b = \frac{R}{\|A_{\lambda_\star}^{-1} v\|} v , \ \rho = \lambda_\star / R.$$

The solution for this instance is unique and satisfies $s_\star^{\text{cr}} = -A_{\lambda_\star}^{-1} b = -R A_{\lambda_\star}^{-1} v / \|A_{\lambda_\star}^{-1} v\|$ so that $\|s_\star^{\text{cr}}\| = R$, and moreover we note that $\|b\| \to 0$ as $\varepsilon \to 0$. For every $s \in \mathcal{K}_t(M, v) = \mathcal{K}_t(A, b)$,

$$\hat{f}_{A,b,\rho}(s) = \frac{1}{2} s^T A s + b^T s + \frac{\rho}{3} \|s\|^3 \geq -\|b\| \|s\| + \frac{1}{2}(\lambda_{\min} + \epsilon_t) \|s\|^2 + \frac{\rho}{3} \|s\|^3 .$$

The RHS above is minimal for

$$\|s\| = \tilde{R} := -\frac{\lambda_{\min} + \epsilon_t}{2\rho} + \sqrt{\left( \frac{\lambda_{\min} + \epsilon_t}{2\rho} \right)^2 + \frac{\|b\|}{\rho}} \leq \frac{-\lambda_{\min} - \epsilon_t}{\rho} + \sqrt{\frac{\|b\|}{\rho}},$$

where the bound holds since our definition of $\epsilon_t$ implies $\epsilon_t \leq -\lambda_{\min}$ and so $-\lambda_{\min} - \epsilon_t \geq 0$. The minimum value of the RHS satisfies

$$\hat{f}_{A,b,\rho}(s) \geq -\frac{2}{3} \|b\| \tilde{R} - \frac{1}{6}(-\lambda_{\min} - \epsilon_t)\tilde{R}^2. \tag{33}$$

Taking without loss of generality $u_{\min}^T b \leq 0$ and using $\rho = \lambda_\star / R$, and $\lambda_\star = -\lambda_{\min} + \varepsilon$, we have

$$\hat{f}_{A,b,\rho}(s_\star^{\text{cr}}) \leq \hat{f}_{A,b,\rho}(R \cdot u_{\min}) \leq \frac{1}{2}\lambda_{\min} R^2 + \frac{1}{3}\lambda_\star R^2 = \frac{1}{6}\lambda_{\min} R^2 + \frac{\varepsilon}{3} R^2. \tag{34}$$

Recall that $\|b\| \to 0$ as $\varepsilon \to 0$, and take $\varepsilon > 0$ sufficiently small so that

$$\varepsilon < \epsilon_t / 24 \ \text{ and } \ \|b\| \leq \min\{\epsilon_t R / 24, \epsilon_t^2 / \rho\},$$

which implies also

$$\tilde{R} \leq \frac{-\lambda_{\min} - \epsilon_t}{\rho} + \frac{\epsilon_t}{\rho} = \frac{-\lambda_{\min}}{\rho} \leq \frac{\lambda_\star}{\rho} = R.$$

Using $\tilde{R} \leq R$, we may replace $\tilde{R}$ with $R$ in the bound (33), and combining this with (34) and the bounds on $\|b\|$ and $\varepsilon$ we obtain

$$\hat{f}_{A,b,\rho}(s) - \hat{f}_{A,b,\rho}(s_\star^{\text{cr}}) \geq \frac{\epsilon_t}{6} R^2 - \frac{2 \|b\|}{3} R - \frac{\varepsilon}{3} R^2 \geq \frac{\epsilon_t}{8} R^2.$$

Recalling $\|s_\star^{\text{cr}}\| = R$ and the definition (32) of $\epsilon_t$, we get the desired "non-convex" lower bound.

To derive the alternative, "convex" lower bound, we again let $0 < \varepsilon < \lambda_{\max} - \lambda_{\min}$ be a parameter to be determined, and we apply Lemma 5 with $n = t$, $\alpha = \varepsilon$, $\beta = \lambda_{\max} - \lambda_{\min}$ to obtain points $\xi_0, \dots, \xi_t \in [0, \lambda_{\max} - \lambda_{\min}]$ and probability masses $\pi_0, \dots, \pi_t$ such that

$$\min_{p \in \mathcal{P}_n} \sum_{k=0}^{n} \pi_k(\xi_k - \varepsilon)(1 - \xi_k p(\xi_k))^2 = \left[ \mathfrak{U}_t\left( \frac{\lambda_{\max} - \lambda_{\min}}{\varepsilon} \right) \right]^2.$$

To construct the hard instance we again set

$$\lambda_\star = -\lambda_{\min} + \varepsilon.$$

Letting $\xi$ and $\sqrt{\pi}$ denote vectors with entries $\xi_i$ and $\sqrt{\pi}_i$, we set

$$A = \operatorname{diag}(\xi - \lambda_\star)\,, \ b = R \cdot A_{\lambda_\star} \sqrt{\pi}\,, \ \rho = \lambda_\star / R.$$

Again we have that $s_\star^{\mathsf{cr}} = -A_{\lambda_\star}^{-1} b$ is the unique solution and $\|s_\star^{\mathsf{cr}}\| = R \|\sqrt{\pi}\| = R$. Let $s \in \mathcal{K}_t(A, b)$, then

$$s = -p(A_{\lambda_\star})b = p(A_{\lambda_\star}) A_{\lambda_\star} s_\star^{\mathsf{cr}} = -Rp(A_{\lambda_\star}) A_{\lambda_\star} \sqrt{\pi}$$

for some $p \in \mathcal{P}_t$. By equality (28) we have

$$\hat{f}_{A,b,\rho}(s) - \hat{f}_{A,b,\rho}(s_\star^{\mathsf{cr}}) \geq \frac{1}{2} \left\| A_{\lambda_\star}^{1/2}(s - s_\star^{\mathsf{cr}}) \right\|^2 = \frac{R^2}{2} \sum_{k=0}^{t} \pi_k \xi_k (1 - \xi_k p(\xi_k))^2$$

$$\geq \frac{R^2}{2} \sum_{k=0}^{t} \pi_k (\xi_k - \varepsilon)(1 - \xi_k p(\xi_k))^2 = \frac{R^2}{2} \left[ \mathfrak{U}_t \left( \frac{\lambda_{\max} - \lambda_{\min}}{\varepsilon} \right) \right]^2.$$

Note that

$$\lim_{\varepsilon \to 0} \mathfrak{U}_t \left( \frac{\lambda_{\max} - \lambda_{\min}}{\varepsilon} \right) = \frac{\sqrt{\lambda_{\max} - \lambda_{\min}}}{2t + 1}.$$

Therefore, we can choose $\varepsilon$ sufficiently small so that

$$\left[ \mathfrak{U}_t \left( \frac{\lambda_{\max} - \lambda_{\min}}{\varepsilon} \right) \right]^2 \geq \frac{\lambda_{\max} - \lambda_{\min}}{2(2t + 1)^2},$$

which gives the proof for the "convex" lower bound, as $\|s_\star^{\mathsf{cr}}\| = R$. □

## E  Numerical experiment details

**Random problem generation, $\kappa < \infty$** We generate random cubic regularization instances $(A, b, \rho)$ as follows. We take $\lambda_{\max} = 1$ and draw $\lambda_{\min} \sim U[-1, -0.1]$, where $U[a, b]$ denotes the uniform distribution on $[a, b]$. We then fix two eigenvalues of $A$ to be $\lambda_{\min}, \lambda_{\max}$ and draw the other $d - 2$ eigenvalues independently from $U[\lambda_{\min}, \lambda_{\max}]$. We then take $A$ to be diagonal with said eigenvalues. This is without much loss of generality (as the Krylov subspace method is rotationally invariant), and it allows us to quickly compute matrix-vector products, whose computation nevertheless accounts for much of the experiment running time when using $d = 10^6$.

For a desired condition number $\kappa$, we let

$$\lambda_\star := \frac{\lambda_{\max} - \kappa \lambda_{\min}}{\kappa - 1}$$

and as usual denote $A_{\lambda_\star} = A + \lambda_\star I$. To generate $b, \rho$, we draw a standard normal $d$-dimensional vector $v \sim \mathcal{N}(0; I)$ and let

$$b = \sqrt{\frac{2}{v^T A_{\lambda_\star}^{-1} v + \frac{\lambda_\star}{3} v^T A_{\lambda_\star}^{-2} v}} \cdot v\,, \ \rho = \frac{\lambda_\star}{\|A_{\lambda_\star}^{-1} b\|},$$

The above choice of $b$ and $\rho$ guarantees that $\rho \|A_{\lambda_\star}^{-1} b\| = \lambda_\star$ and therefore $s_\star^{\mathsf{cr}} = -A_{\lambda_\star}^{-1} b$ is the unique solution and the problem condition number satisfies

$$\frac{\lambda_{\max} + \rho \|s_\star^{\mathsf{cr}}\|}{\lambda_{\min} + \rho \|s_\star^{\mathsf{cr}}\|} = \frac{\lambda_{\max} + \lambda_\star}{\lambda_{\min} + \lambda_\star} = \kappa$$

as desired. Moreover, our scaling of $b$ guarantees that

$$\hat{f}_{A,b,\rho}(0) - \hat{f}_{A,b,\rho}(s_\star^{\mathsf{cr}}) = \frac{1}{2}(s_\star^{\mathsf{cr}})^T A_{\lambda_\star} s_\star^{\mathsf{cr}} + \frac{\rho}{6} \|s_\star^{\mathsf{cr}}\|^3 = \frac{1}{2} \left( b^T A_{\lambda_\star}^{-1} b + \frac{\lambda_\star}{3} b^T A_{\lambda_\star}^{-2} b \right) = 1.$$

Our technique for generating $(A, b, \rho)$ is similar to the one we used in [5] to test gradient descent for cubic regularization. The main difference is that in [5] the value of $\rho$ is fixed and consequently there is no control over the initial optimality gap.

For every value of $\kappa$, we generate 5,000 problem instances independently as described above.

Figure 2: Optimality gap of Krylov subspace solutions on random cubic-regularization problems, versus subspace dimension $t$. Each plot shows result for problem instances with a different eigengap $\gamma = (\lambda_{\max} - \lambda_{\min})/(\lambda_2 - \lambda_{\min})$, where $\lambda_2$ is the smallest eigenvalue larger than $\lambda_{\min}$. Each line represents median suboptimality, and shaded regions represent inter-quartile range. Different lines correspond to different randomization settings.

**Random problem generation,** $\kappa = \infty$   We let $A = \operatorname{diag}(\lambda)$ where $\lambda_1 = \lambda_{\min} = -0.5$, $\lambda_d = \lambda_{\max} = 0.5$ and $\lambda_2, \ldots, \lambda_{d-1}$ are drawn i.i.d. from $U[\lambda_{\min} + \gamma, \lambda_{\max}]$ where we take the eigen-gap $\gamma = 10^{-4}$ and $d = 10^6$.

As $\kappa = \infty$, we let

$$\lambda_\star = -\lambda_{\min}$$

and denote $\hat{A}_{\lambda_\star} \coloneqq \operatorname{diag}(\lambda_2 + \lambda_\star, \ldots, \lambda_{\max} + \lambda_\star)$. We generate $b$ and $\rho$ by drawing a standard normal $(d-1)$-dimensional vector $v$, and letting

$$b_1 = 0 \,, \; b_{2:d} = \sqrt{\frac{2}{v^T \hat{A}_{\lambda_\star}^{-1} v + (1 + \tau^2)\frac{\lambda_\star}{3} v^T \hat{A}_{\lambda_\star}^{-2} v}} \, v \,, \; \rho = \frac{\lambda_\star}{\|\hat{A}_{\lambda_\star}^{-1} b_{2:d}\| \sqrt{1 + \tau^2}},$$

where $\tau$ is a parameter that determines the weight of the eigenvector corresponding to $\lambda_{\min}$ in the solution (when $\tau = \infty$ we have a pure eigenvector instance); we take $\tau = 10$. A global minimizer $s_\star^{\mathsf{cr}}$ of the problem instance $(A, b, \rho)$ generated above has the form,

$$[s_\star^{\mathsf{cr}}]_1 = \pm \tau \|\hat{A}_{\lambda_\star}^{-1} b_{2:d}\| \,, \; [s_\star^{\mathsf{cr}}]_{2:d} = -\hat{A}_{\lambda_\star}^{-1} b_{2:d}.$$

As in the case $\kappa < \infty$, it is easy to verify that the scaling of $b$ guarantees $\hat{f}_{A,b,\rho}(0) - \hat{f}_{A,b,\rho}(s_\star^{\mathsf{cr}}) = 1$.

When $\kappa = \infty$, the choice of eigen-gap $\gamma$ strongly affects optimization performance. We explore this in Figure 2, which repeats the experiment described above with different values of $\gamma$ (and $d = 10^5$). As seen in the figure, the non-randomized Krylov subspace solution becomes more suboptimal as $\gamma$ increases. Moreover, randomization "kicks-in" after roughly $\log d / \sqrt{\gamma}$ iterations, when eigen-gap-dependent linear convergence begins.

To create each plot, we draw 10 independent problem instances from the distribution described above, and for each problem instance run each randomization approach with 50 different random seeds; we observe that sampling problem instances and sampling randomization seeds contribute similar amount of variation to the final ensemble of results.

**Hardness of generated problems**   It is well known that the performance of subspace methods improves dramatically when the eigenvalues of $A$ are clustered [35]. Taking the eigenvalues of $A$ to be uniformly distributed produces very little clustering, making the instances we draw somewhat hard. However, examining the proof of the lower bound (19) we see that the worst case eigenvalues are of the form $\lambda_k = \lambda_{\min} + (\lambda_{\max} - \lambda_{\min}) \sin^2 \theta_k$ where $\theta_1, \ldots \theta_d$ are equally spaced in $[0, \pi/2]$. This is fairly different from a uniform distribution (asymptotically as $d \to \infty$ it becomes an arcsine distribution), and consequently we think that uniformly distributing the eigenvalues makes for a challenging but not quite adversarial test case.

**Computing Krylov subspace solutions**  We use the Lanczos process to obtain a tridiagonal representation of $A$ as described in Section A. To obtain full optimization traces we solve equation (23) after every Lanczos iteration, warm-starting $\lambda$ with the solution from the previous step and the minimum eigenvalue of the current tridiagonal matrix. We use the Newton method described by Cartis et al. [9, Algorithm 6.1] to solve the equation (23) in the Krylov subspace. For the $\kappa < \infty$ experiment, we stop the process when $\left| \left\| A_\lambda^{-1} b \right\| - \lambda/\rho \right| < 10^{-12}$ or after 25 tridiagonal system solves are computed. For the $\kappa = \infty$ experiment we allow up to 100 system solves.

## Footnotes

[2] translating to the notation of [37], take $\phi(x, v) = g(x)$ and $P(x)$ to be the indicator of $Q$, so that $q^P(\cdot) = g(x^\star)$, note that $\theta_k$ ($\alpha_k$ in our notation) satisfies $\theta_k \leq 2/(2+k)$. We discuss alternative references for this result after the proof.