[Reviews · NeurIPS 2018]

Reviewer 1



This paper gives a complete analysis of how many iterations are required for a Krylov subspace method to approximately solve the “trust region” and “cubic-regularized” quadratic minimization problems. These problems take the form: min x^T A x + b^Tx subject to ||x|| <= R or with an additional regularization term of + param*||x||^3. A is a symmetric, but not necessarily PSD matrix (i.e. it can have negative eigenvalues). The objective function is not necessarily convex. Problems of this form are important in a number of applications, especially as subroutines for regularized Newton methods. In addition to their practical importance, such methods have recently been used to give the fastest theoretical runtimes for finding stationary points and local minima of general non-convex objective functions. The key ingredient in this recent work are provably fast approximation algorithms for the cubic-regularized quadratic minimization problem. The goal of this paper is to analyze a significantly simpler approach for solving both regularized minimization problems approximately. In particular, either problem can be solved via a standard Krylov subspace method: compute b, Ab, A^2b, ..., A^tb for some relative small parameter t and optimize the objective function subject to the constraint that the solution lies in the span of these vectors. This can be done efficiently, although I would like to see a worst-case theoretical runtime stated for the subproblem (even if it’s slower than using Newton’s method as the authors suggest). Regardless, in most cases, the dominant cost of the algorithm is the computation of b, Ab, A^2b, …, A^tb. The author’s main contribution is provide upper bounds (and nearly matching lower bounds) on how many iterations t are required to obtain a good approximation solution. They provide bounds that depend polynomially on a natural condition number of the optimization problem and logarithmically on the desired accuracy epsilon, as well as bounds that depend polynomially on epsilon, but hold even for poorly conditioned problems. These later bounds show that the basic Krylov method matches what was established in previous work using more complex algorithms. The main technical tools used are standard bounds from polynomial approximation theory, which the authors first use to analyze the trust region problem (with ||x|| constrained to be < R). This result then immediately yields a result for the cubic regularization problem, since an optimal solution to that problem is teh same as an optimal solution to the trust region problem with a specific setting of R. While the analysis is relatively straightforward and should not surprise those familiar with Krylov methods, the convergence results obtained haven’t appeared in the literature before (even while more complex algorithms have been studied). For me, this makes the paper is a clear accept. It is also well written and easy to follow and I appreciated the matching lower bounds. Small comments: - Is it possible to state a worst-case runtime bound for solving the problem restricted to the Krylov subspace using Newton’s method? I think it would be nice to include. Alternatively, is it clear how to solve the problem using a slower direct method (i.e. eigendecomp of the tridiag Lancoz matrix)? If so, what’s the runtime? - I like that most of the proofs pretty directly rely on basic polynomial approximation bounds. Is there some reason these technique don’t work for Lemma 2? It seems to me that it should be possible to prove directly, instead of through Tseng’s version of accelerated gradient descent. In particular, using your Lemma 4, it should be possible to directly argue that after t = \tilde{O}(sqrt(lambda_max/eps)) iterations, there is some x in the Krylov subspace such that ||x - s_*|| <= (epsilon/\lambda_max)*||s_*||. Then the claim would follow from the fact that x^TAx - s_*^TAs_* <= epsilon*||s_*|| and b^T(s_* - x) <= eps/lambda_max*||s_*|| ||b|| <= epsilon*||s_*||^2. Is the problem that this approach would lose a log factor? If so, can it be improved by using the standard accelerated gradient polynomial, instead of appealing to Tseng’s version? E.g. the polynomial from https://blogs.princeton.edu/imabandit/2013/04/01/acceleratedgradientdescent/? I find the reliance on Tseng’s analysis makes the paper feel somewhat incomplete since is seems like the reference work has never appeared in publication, nor been cited (correct me if this is wrong), so presumably it is not particularly well vetted. - At line 585 if seems like a reference is missing.

Reviewer 2



This work provides a convergence rate for Krylov subspace solutions to the trust-region and cubic-regularization subproblems, which is of rising interest in solving nonconvex, nonlinear optimization problems recently. The paper is well organized and well-written overall. Below are some detailed comments. 1. Linear rate vs sublinear rate: for the general case (vs ‘hard case’) of subproblems, the authors provide two convergence rate, one is linear, and another sublinear. It seems like the convergence behavior is similar to the tail behavior of sub-exponential r.v., which has two different behaviors when t varies. Here, for better presentation of the result, it is better to present the convergence rates in a similar favor of tail behavior of sub-exponential r.v., that convergence rate of the Krylov subspace solution behaves like sublinear when t \ll \sqrt{kappa}, and linear when t >= \sqrt{kappa}, and make this point clear early in the remark of the result rather than in the experiment part. Otherwise, it is a little bit confusing why the author presents two convergence rate for the same algorithm, for which the linear rate is obviously better than sublinear for large t. 2. ”Hard case": "hard case" is one of the major challenges for solving the trust-region and cubic regularization subproblems. The general analysis of this work does not apply to the hard case’’ neither, and this work proposed a randomization approach to deal with hard case’’. With high probability, the work provides a sublinear convergence rate for the randomization approach. Since it is almost impossible to check the hard case a-priori, how does the proposed method work in practice? For running the trust-region method, do we need to do randomization for each subproblem? If so, does it imply that the convergence rate of Krylov subspace solutions is sublinear in practice? Does it possible to provide a linear rate result for the "hard case" using random perturbations or is there some fundamental challenge for the "hard case”? 3. For the experiment part, the authors need to show some experiments of the convergence behavior of the randomizing approach of the Krylov subspace solutions for the "hard cases".

Reviewer 3



Summary The authors consider optimization problems where one tries to minimize a (possibly non-convex) quadratic function in two different settings: - when the optimization domain is a ball with fixed radius, - when a cubic regularizer is added to the quadratic function. These two settings are motivated by their importance for second-order optimization. The goal of the article is to describe the convergence rate, in these settings, of Krylov subspace solutions (the Krylov subspace solution, at order t, is the solution when the problem is restricted to a t-dimensional subspace naturally associated with the quadratic function). The authors establish two convergence rates: one that is exponential in t, with a rate that depends of the (properly defined) condition number of the problem, and another one that scales like t^(-2), with a constant that depends logarithmically on the scalar product between two specific vectors (and that scales at worst like log^2(d) when a random perturbation is added). They show that the two bounds are essentially tight. Main remarks I am unfortunately not very familiar with Krylov subspace methods and second-order optimization, so it is difficult for me to evaluate the position of this article in the literature, and its importance to the domain. Nevertheless, as a non-expert, I can say that I sincerely enjoyed reading this article, and found it very interesting; I hope it will be published. In more detail, - Krylov subspace methods are an extremely important family of optimization algorithms. I therefore think that it is very valuable to precisely understand which convergence guarantees they can offer in all important use cases, and to develop new tools for proving such guarantees (as well as to improve the previous tools). The motivations from trust-regions and cubic-regularized Newton methods are also important. - The article is very well-written, especially the technical parts. The proofs rely on polynomial approximation results that are relatively classical; the technical novelty lies in how these results are used, and combined with previous literature, to obtain the two convergence rates. The classical results are presented in full detail in the appendix; with this respect, I appreciated the effort that has been put into making the article self-contained. The rest is very clearly explained in the main part of the article, with all the necessary details, but also enough discussion so that the reader can get a good intuition of the principles behind the proofs. - The two convergence rates, and the fact that they are optimal, are essentially the same as the optimal convergence rates for solving convex problems with first-order methods in two situations : (i) when the convex function is assumed to have a Lipschitz gradient (in which case the rate is O(t^(-2))) (ii) when the convex function is strongly convex, and has a Lipschitz gradient (in which case the rate is exponential, with the same dependency in the condition number as in Equation (4) of this article). Maybe the link could be discussed and clarified in the article? Minor remarks and typos - Line 38: I did not find the expression "the sub-optimality ... is bounded by" very clear. - Line 120: "is choose" -> "is to choose"? - Line 220: "and os it is" -> "and it is" - Section 4, first paragraph: This is a minor point but, since the gap is smaller in the case of the cubic-regularized problem, is it immediate how to deduce the trust-region result from the cubic-regularized one, including the replacement of the cubic gap by the trust-region one in the right-hand side of Equation (18)? - Line 238: tau has to be larger than 1. - Bibliography: Reference [14] contains some weird characters, and it seems to me that the link given for reference [32] is broken. - Line 402: "for all y>=0" and "for all z>=1" could be added. - Line 415: in the left-hand side of the double inequality, I think that sqrt(kappa) should be sqrt(kappa)-1. - Line 417, second line of the block: "n-1" should be "n+1". - Line 440: g must be convex. - Line 485: I think that the last lambda_max should be a lambda_min. - The equation after line 489 could be split into two lines. - Line 491: The definition of n is missing. - Line 496: "(t+1)-dimensional" -> "d-dimensional"? - Equation after line 496: The definition of rho is actually the definition of 1/rho (we want lambda_star = rho ||s_star||). - Equation after line 501: There is a Delta missing. - Equaiton after line 525, left-hand side: "q^2(0)(u^Tv)" is probably missing in the denominator. - Equation after line 554: I do not get why the first R is not a \tilde R. - Equation after line 562: I think there is a minus sign missing in the right-hand side. - Line 578: "guarantee" -> "guarantees" Added after author feedback Thanks to the authors for taking my minor remarks into account. I am naturally still in favor of accepting the article.